# BM$^2$: Coupled Schrödinger Bridge Matching

**Stefano Peluchetti**                                                                 *stepelu@sakana.ai*
*Sakana AI*

**Reviewed on OpenReview:** *https://openreview.net/forum?id=fqkq1MgONB*

## Abstract

A Schrödinger bridge establishes a dynamic transport map between two target distributions via a reference process, simultaneously solving an associated entropic optimal transport problem. We consider the setting where samples from the target distributions are available, and the reference diffusion process admits tractable dynamics. We thus introduce Coupled Bridge Matching (BM²), a simple *non-iterative* approach for learning Schrödinger bridges with neural networks. A preliminary theoretical analysis of the convergence properties of BM² is carried out, supported by numerical experiments that demonstrate the effectiveness of our proposal.

## 1 Introduction

The Schrödinger bridge problem seeks a process, the Schrödinger bridge, with prescribed initial and terminal distributions, such that the distribution of the Schrödinger bridge minimizes the Kullback-Leibler (KL) divergence to the distribution of a reference process. Schrödinger bridges play a central role in measure transport theory (Marzouk et al., 2016). Notably, it is known that the initial-terminal distribution of a Schrödinger bridge provides a solution to a corresponding entropic optimal transport problem (Peyré & Cuturi, 2020). Schrödinger bridges thus provide an effective framework for finding an alignment between samples from two target distributions. Furthermore, diffusion-based generative models (Ho et al., 2020; Song et al., 2021) can be interpreted as solving trivial instances of the Schrödinger bridge problem (Peluchetti, 2023). Consequently, Schrödinger bridges offer a more general approach to contemporary generative applications.

We consider the setting where samples are readily available from both target distributions, and where the reference process is a diffusion process solution to a stochastic differential equation (SDE). We thus introduce Coupled Bridge Matching (BM²), a novel methodology aimed at computing the Schrödinger bridge given the reference SDE and samples from the two marginal distributions of interest. BM² builds upon Bridge Matching (BM), introduced[1] by Peluchetti (2021). Our approach advances recent contributions by Peluchetti (2023); Shi et al. (2023) by removing the need to solve a sequence of optimization problems. A neural network is employed to jointly learn a forward drift function and a backward drift function corresponding to the forward and backward dynamics of a Schrödinger bridge. BM² achieves several key desiderata:

(i) non-iterative: training is conducted through standard stochastic gradient descent within a single optimization loop;

(ii) exact: the idealized version of BM² yields the target Schrödinger bridge without approximations; the only sources of error involved in its practical implementation are the neural network approximation error and the discretization error due to sampling the learned SDE;

(iii) efficient gradient: the gradient of the loss function with respect to neural network parameters depends solely on few random variables sampled at the current optimization step;

---

[1] Peluchetti (2021) used the term "Diffusion Bridge Mixture-Matching Transport" (DBMT), but we follow Shi et al. (2023) in using the sleeker nomenclature "Bridge Matching" for this transport.

(iv) simple loss: the loss function avoids derivative terms with respect to neural network inputs and does not impose hard constraints (such as conservative vector field requirements) on the neural network approximator.

These features collectively enhance the efficiency and applicability of BM$^2$ in solving Schrödinger bridge problems. Training is robust, as it does not depend on hyperparameters that are typically challenging to set without time-consuming pilot runs, such as the number of training steps per optimization iteration (i) or the level of approximation (ii). Moreover, the memory requirements are modest due to (iii). Finally, the implementation is straightforward (i, iv), as illustrated in Algorithms 1 and 2 and in the annotated PyTorch code of Listing 1.

**Content**: This paper is structured as follows. In Section 2, we formally introduce the Schrödinger bridge problem with associated reference process dynamics. Section 3 reviews Bridge Matching, while Section 4 introduces Coupled Bridge Matching, discussing its theoretical properties and implementation aspects. Numerical experiments are presented in Section 5, followed by a discussion of related works in Section 6. Section 7 concludes the paper. For clarity, a more general formulation of BM$^2$ is deferred to Appendix A, all proofs to Appendix B, an additional numerical experiment to Appendix C, and code listings to Appendix D.

**Notation and Assumptions**: To enhance accessibility, we refrain from discussing the more technical aspects related to the Schrödinger bridge problem in its path measure formulation. The excellent treaties of Léonard (2014b;a) and Bortoli et al. (2021, Appendices D, H) already serve this goal. We denote distributions with uppercase letters and their corresponding (Lebesgue) densities with lowercase letters. All stochastic processes considered are $d$-dimensional, continuous, and defined on the unit time interval $[0, 1]$. For a stochastic process $X$ with distribution $P$ (denoted $X \sim P$), we use subscripts to specify marginal distributions, joint distributions, and conditional distributions of $P$. $P_t$: marginal distribution of $X_t$ at time $t$, with density $p_t$; $P_{0,1}$: initial-terminal joint distribution of $(X_0, X_1)$; $P_{|0}$: distribution of $X$ given its initial value $X_0$. Superscripts indicate a distribution $P$'s dependency on another distribution $Z$, as in $P^Z$, or a sequence of distributions, as in $P^{(i)}$, $i \geq 1$. For a $d$-dimensional distribution $Q_0$, we define the stochastic process mixture distribution $Q_0 P_{|0}$ as: $(Q_0 P_{|0})(X \in \cdot) := \int P_{|0}(X \in \cdot | x_0) Q_0(dx_0)$. From a generative perspective, $X \sim Q_0 P_{|0}$ is obtained by sampling $X_0 \sim Q_0$ and then $X \sim P_{|0}(\cdot | X_0)$ conditionally on $X_0$. The marginal-conditional decomposition of $P$ over its initial value is thus $P = P_0 P_{|0}$. Similarly, for a $d \times d$-dimensional joint distribution $Q_{0,1}$, we define $Q_{0,1} P_{|0,1}$ such that $X \sim Q_{0,1} P_{|0,1}$ is obtained by sampling $(X_0, X_1) \sim Q_{0,1}$ and then $X \sim P_{|0,1}(\cdot | X_0, X_1)$ conditionally on $X_0$ and $X_1$. Time is always indexed on a common forward timescale, on which all stochastic processes' distributions are defined. The dynamics of a diffusion process $X \sim P$ can be formulated in both forward and backward time directions, through corresponding forward and backward SDEs. In backward SDEs, $t$ decreases from 1 to 0 ($dt$ is negative), which is denoted by $t \in [1, 0]$. All Brownian motions are independent. Unless otherwise noted, each diffusion process is a Markov diffusion process which is a (weak) solution to an associated SDE.

## 2 Problem Setting

### 2.1 Schrödinger Bridges and Entropic Optimal Transport

For two target $d$-dimensional distributions $\Psi_0$ and $\Psi_1$, and a reference stochastic process distribution $R$, the *dynamic* Schrödinger bridge (SB) problem seeks to find

$$S^{\Psi_0, \Psi_1, R} := \operatorname*{arg\,min}_{P \in \mathcal{P}(\Psi_0, \Psi_1)} \mathbb{KL}(P \parallel R), \tag{1}$$

where $\mathbb{KL}(\cdot \parallel \cdot)$ is the KL divergence and $\mathcal{P}(\Psi_0, \Psi_1)$ is the class of distributions of stochastic processes having initial distribution $\Psi_0$ and terminal distribution $\Psi_1$. We narrow down (1) to the case where $R$ is the distribution of a diffusion process. In this case, under suitable conditions (Léonard, 2014b), (1) admits a unique solution which is also a diffusion process. From this point forward, $\Psi_0$, $\Psi_1$ and $R$ are considered fixed. For brevity, we will thus denote the Schrödinger bridge $S^{\Psi_0, \Psi_1, R}$ simply as $S$, and apply the same notation convention to any distribution dependent on these variables.

The forward and backward dynamics of $X \sim S$ are given by:

$$X_0 \sim \Psi_0, \quad dX_t = \mu_s(X_t, t)dt + \sigma dW_t, \quad t \in [0,1], \tag{$S$}$$

$$X_1 \sim \Psi_1, \quad dX_t = -\upsilon_s(X_t, t)dt + \sigma dW_t, \quad t \in [1,0], \tag{$\overleftarrow{S}$}$$

for the SB-optimal drift functions $\mu_s, \upsilon_s$. These functions are related to the Schrödinger potentials (Léonard, 2014b) and are not analytically available aside from very specific choices of $\Psi_0, \Psi_1$ and $R$.

We assume that $R_{0,1}$ admits density $r_{0,1}$. Once $S$ is obtained, the solution to the *static* Schrödinger bridge problem is given by $S_{0,1}$:

$$
\begin{aligned}
S_{0,1} &= \underset{C_{0,1} \in \mathcal{C}(\Psi_0, \Psi_1)}{\arg\min} \ \mathbb{KL}(C_{0,1} \parallel R_{0,1}), \\
&= \underset{C_{0,1} \in \mathcal{C}(\Psi_0, \Psi_1)}{\arg\min} \ \underset{C_{0,1}}{\mathbb{E}} \left[ -\log r_{1|0}(X_1|X_0) \right] - \mathbb{H}(C_{0,1}).
\end{aligned}
\tag{2}
$$

In (2), $\mathcal{C}(\Psi_0, \Psi_1)$ denotes the class of $d \times d$-dimensional joint distributions with marginal distributions $\Psi_0$ and $\Psi_1$, commonly referred to as the class of couplings of $\Psi_0$ and $\Psi_1$, and $\mathbb{H}(C_{0,1}) := \mathbb{E}_{C_{0,1}}[-\log c_{1,0}(X_1, X_0)]$ is the entropy of $C_{0,1}$.

The entropic optimal transport (EOT) solution for the cost function $k(x_0, x_1)$ and regularization level $\varepsilon$ is given by:

$$E_{0,1} := \underset{C_{0,1} \in \mathcal{C}(\Psi_0, \Psi_1)}{\arg\min} \ \underset{C_{0,1}}{\mathbb{E}} \left[ \kappa(X_1, X_0) \right] - \varepsilon \mathbb{H}(C_{0,1}). \tag{3}$$

Thus, for each choice of $R_{0,1}$ in (2), $S_{0,1}$ solves a corresponding problem (3). As in the following, when $R$ is associated to $(R)$, $S_{0,1}$ solves the EOT problem (3) for the Euclidean cost $\kappa(x_1, x_0) = \frac{1}{2}\|x_0 - x_1\|^2$ and regularization level $\varepsilon = \sigma^2$.

We refer to Peyré & Cuturi (2020); Léonard (2014b); Gushchin et al. (2023) for related background material from complementary perspectives.

## 2.2 Reference Dynamics

We focus on the case where $R$ is the distribution of a scaled Brownian motion:

$$X_0 \sim \Psi_0, \quad dX_t = \sigma dW_t, \quad t \in [0,1], \tag{$R$}$$

with $\sigma > 0$. Our approach *is not limited* to the choice of SDE $(R)$, $\text{BM}^2$ readily extends to the broader class of reference SDEs examined in Peluchetti (2023). The main requirement for the applicability of $\text{BM}^2$ is the analytical availability of (4, 5) for the chosen reference SDE. We address the case, commonly employed in generative applications, of $dX_t = \sigma\sqrt{\beta_t}dW_t$ for a schedule $\beta_t$ explicitly in Appendix A, and refer the reader to Peluchetti (2021; 2023) for the general setting. As our developments are orthogonal to the specific choice of reference process, we focus on the simplest case for explanatory reasons.

We collect here various results concerning $(R)$ that will be utilized in the following:

$$R_{t|0}(\cdot|x_0) = \mathcal{N}(x_0, \sigma^2 t), \tag{4}$$

$$R_{t|0,1}(\cdot|x_0, x_1) = \mathcal{N}(x_0(1-t) + x_1 t, \sigma^2 t(1-t)), \tag{5}$$

$$\mu_{01}(x_t, t, x_1) := \sigma^2 \nabla_{x_t} \log r_{1|t}(x_1|x_t) = \frac{x_1 - x_t}{1-t}, \tag{6}$$

$$\upsilon_{01}(x_t, t, x_0) := \sigma^2 \nabla_{x_t} \log r_{t|0}(x_t|x_0) = \frac{x_0 - x_t}{t}, \tag{7}$$

$$\gamma_{01}(x_t, t, x_0, x_1) := \sigma^2 \nabla_{x_t} \log r_{t|0,1}(x_t|x_0, x_1) = \frac{x_0(1-t) + x_1 t - x_t}{t(1-t)}. \tag{8}$$

Conditioning $X \sim R$ on the endpoints $X_0 = x_0$, $X_1 = x_1$ results in the diffusion bridge distribution $R_{|0,1}$, with associated forward and backward SDEs:

$$X_0 = x_0, \quad dX_t = \mu_{01}(X_t, t, x_1)dt + \sigma dW_t, \quad t \in [0,1], \tag{$R_{|0,1}$}$$

$$X_1 = x_1, \quad dX_t = -v_{01}(X_t, t, x_0)dt + \sigma dW_t, \quad t \in [1, 0]. \tag{$\overleftarrow{R_{|0,1}}$}$$

## 3 Bridge Matching (BM)

We succinctly review Bridge Matching, and refer to Peluchetti (2021; 2023); Shi et al. (2023) for more details. BM takes as input a joint distribution $Q_{0,1}$ with marginal distributions $Q_0, Q_1$ and a SDE, ($R$). Firstly, a stochastic process $\Pi^{Q_{0,1}}$ is constructed as a mixture of diffusion bridges ($R_{|0,1}$), such that the endpoints $(X_0, X_1)$ of $X \sim \Pi^{Q_{0,1}}$ are distributed according to $Q_{0,1}$. This process, which is a mixture of diffusion processes, is not itself a diffusion process in general (Jamison, 1974). However, we can obtain a marginal-matching diffusion process with distribution $M^{Q_{0,1}}$ for which $M_t^{Q_{0,1}} = \Pi_t^{Q_{0,1}}, 0 \le t \le 1$. Consequently, $X \sim M^{Q_{0,1}}$ is a diffusion process for which $X_0 \sim Q_0$ and $X_1 \sim Q_1$, i.e. it defines a dynamic transport from $Q_0$ to $Q_1$.

Concretely, let $\Pi^{Q_{0,1}} \coloneqq Q_{0,1}R_{|0,1}$. The BM transport based on $Q_{0,1}$ with distribution $M^{Q_{0,1}}$ is realized by

$$X_0 \sim Q_0, \quad dX_t = \underbrace{\mu_m^{Q_{0,1}}(X_t, t)}_{\mathbb{E}_{\Pi^{Q_{0,1}}}[\mu_{01}(X_t, t, X_1)|X_t]}dt + \sigma dW_t, \quad t \in [0, 1], \tag{$M$}$$

$$X_1 \sim Q_1, \quad dX_t = -\underbrace{v_m^{Q_{0,1}}(X_t, t)}_{\mathbb{E}_{\Pi^{Q_{0,1}}}[v_{01}(X_t, t, X_0)|X_t]}dt + \sigma dW_t, \quad t \in [1, 0], \tag{$\overleftarrow{M}$}$$

and satisfies $M_t^{Q_{0,1}} = \Pi_t^{Q_{0,1}}, 0 \le t \le 1$.

As conditional expectations are mean squared error minimizers, suitable training objectives for the drift functions $\mu_m^{Q_{0,1}}$ and $v_m^{Q_{0,1}}$ are derived from

$$\mu_m^{Q_{0,1}} = \arg\min_\mu \mathbb{E}_{\Pi^{Q_{0,1}}} \left[ \frac{1}{2} \int_0^1 \|\mu_{01}(X_t, t, X_1) - \mu(X_t, t)\|^2 dt \right], \tag{9}$$

$$v_m^{Q_{0,1}} = \arg\min_v \mathbb{E}_{\Pi^{Q_{0,1}}} \left[ \frac{1}{2} \int_0^1 \|v_{01}(X_t, t, X_0) - v(X_t, t)\|^2 dt \right], \tag{10}$$

by replacing each integral with an expectation over uniform time $t \sim \mathcal{U}(0, 1)$, and then approximating both expectations with Monte Carlo estimators. While we will rely exclusively on (9, 10) in the experiments of Section 5, $\mu_m^{Q_{0,1}}$ and $v_m^{Q_{0,1}}$ can be inferred from paths $X \sim \Pi^{Q_{0,1}}$ also by performing maximum likelihood estimation or by employing a drift matching estimator (Liu et al., 2022; Peluchetti, 2023).

We conclude this section by reviewing prior BM results relevant for $BM^2$. Define:

$$\mathcal{P} \coloneqq \{d\text{-dimensional, continuous, stochastic processes on } [0, 1]\},$$
$$\mathcal{R} \coloneqq \{P \in \mathcal{P} \mid P = P_{0,1}R_{|0,1} = \Pi^{P_{0,1}} \text{ for some } P_{0,1}\},$$
$$\mathcal{M} \coloneqq \{P \in \mathcal{P} \mid P \text{ is a (Markov) diffusion process}\},$$
$$\mathcal{S} \coloneqq \{P \in \mathcal{P} \mid P \text{ is a Schrödinger bridge for some target marginal distributions}\} = \mathcal{R} \cap \mathcal{M},$$

where the equivalence is established by Jamison (1975) under appropriate assumptions. We additionally define the following restrictions: $\mathcal{P}(\Psi_0, \cdot) \coloneqq \{P \in \mathcal{P} \mid P_0 = \Psi_0\}$, $\mathcal{P}(\cdot, \Psi_1) \coloneqq \{P \in \mathcal{P} \mid P_1 = \Psi_1\}$, $\mathcal{P}(\Psi_0, \Psi_1) \coloneqq \{P \in \mathcal{P} \mid P_0 = \Psi_0 \text{ and } P_1 = \Psi_1\}$. Restrictions to $\mathcal{R}, \mathcal{M}, \mathcal{S}$ and $\mathcal{C}$ employ the same notation.

For $Q \in \mathcal{P}$, it is instructive to view BM as a map between distributions defined by the composition of two projections: $Q \xrightarrow{\mathcal{R}p} \Pi^{Q_{0,1}} \xrightarrow{\mathcal{M}p} M^{Q_{0,1}}$. Here, the *reciprocal projection* $\mathcal{R}p : \mathcal{P} \to \mathcal{R}$ projects $Q$ onto the reciprocal class $\mathcal{R}$, while the *Markovian projection* $\mathcal{M}p : \mathcal{R} \to \mathcal{M}$ projects $\Pi^{Q_{0,1}}$ onto the class of diffusion processes, see Shi et al. (2023). It follows that if $P \in \mathcal{R}$, then $P = \mathcal{R}p(P)$, and if $P \in \mathcal{M}$, then $P = \mathcal{M}p(P)$. Consequently, if $P \in \mathcal{S}$, $P = (\mathcal{M}p \circ \mathcal{R}p)(P)$ for the BM map $(\mathcal{M}p \circ \mathcal{R}p)$, and conversely if $P = (\mathcal{M}p \circ \mathcal{R}p)(P)$ then $P \in \mathcal{S}$.

### 3.1 Iterated Bridge Matching (I-BM) and Diffusion Iterative Proportional Fitting (DIPF)

In the dynamic setting, Peluchetti (2023); Shi et al. (2023) demonstrate that, under suitable conditions, iterative application of the BM procedure to an initial coupling $C_{0,1} \in \mathcal{C}(\Psi_0, \Psi_1)$ results in convergence

toward $S$. Specifically, defining $I^{(0)} := M^{C_{0,1}}$ and $I^{(i)} := M^{I_{0,1}^{(i-1)}}$ for $i \geq 1$, it holds that $\mathbb{KL}(I^{(i)} \parallel S) \to 0$ as $i \to \infty$. In practical applications, the independent initial coupling given by the product distribution $C_{0,1} = \Psi_0 \otimes \Psi_1$ is frequently employed.

In the static setting, the classical procedure employed in solving problems (2, 3) is known by several names: the Sinkhorn algorithm (Peyré & Cuturi, 2020), the Iterated Proportional Fitting (IPF) procedure (Ruschendorf, 1995), or Fortet iterations (Fortet, 1940). The iterates are given by $D_{0,1}^{(0)} := \Psi_0 R_{1|0}$, $D_{0,1}^{(1)} := \Psi_1 K_{0|1}^{(0)}$, $D_{0,1}^{(2)} := \Psi_0 D_{1|0}^{(1)}$, and so on. At each iteration, one of the target marginal distributions is replaced while the remaining conditional distribution is kept fixed. Alternatively, one can start from $\Psi_1 R_{0|1}$ following the same logic. Under suitable conditions (Ruschendorf, 1995), KL convergence $\mathbb{KL}(S_{0,1} \parallel D_{0,1}^{(i)}) \to 0$ is established. The key insight of Bortoli et al. (2021); Vargas et al. (2021) is that it is possible to extend the IPF iterations to the dynamic setting. In this case, the IPF iterations are implemented by learning the time reversal of a diffusion process at each iteration. We refer to the resulting training algorithm, as proposed by Bortoli et al. (2021), as Diffusion Iterative Proportional Fitting (DIPF). Bortoli et al. (2021) establishes the convergence properties of the DIPF iterates, see their Propositions 4, 5 and Section 3.5.

## 4 Coupled BM (BM$^2$)

As a starting point for the derivation of BM$^2$, consider the system of equations

$$\begin{cases} H^{K'_{0,1}} = \Psi_0 M_{|0}^{K'_{0,1}} \\ K^{H'_{0,1}} = \Psi_1 M_{|1}^{H'_{0,1}} \end{cases}, \tag{11}$$

whose variables are diffusion distributions $H^{K'_{0,1}}, K^{H'_{0,1}}$ and $H', K'$. That is, $H^{K'_{0,1}}$ is obtained as the BM transport based on $K'_{0,1}$ conditioned to have initial distribution $\Psi_0$, while $K^{H'_{0,1}}$ is obtained as the BM transport based on $H'_{0,1}$ conditioned to have terminal distribution $\Psi_1$. Equivalently, (11) is expressed as

$$X_0 \sim \Psi_0, \quad dX_t = \mu_m^{K'_{0,1}}(X_t, t)dt + \sigma dW_t, \quad t \in [0,1], \tag{$H^{K'_{0,1}}$}$$

$$X_1 \sim \Psi_1, \quad dX_t = -\upsilon_m^{H'_{0,1}}(X_t, t)dt + \sigma dW_t, \quad t \in [1,0]. \tag{$\overleftarrow{K}^{H'_{0,1}}$}$$

All of $\mu_m, \upsilon_m, M$ are defined in Section 3. System (11) defines an update step $(H', K') \overset{(11)}{\mapsto} (H^{K'_{0,1}}, K^{H'_{0,1}})$. We are interested in the fixed points of such updates, i.e. $(H', K')$ such that $(H', K') \overset{(11)}{\mapsto} (H', K')$. It holds that $H' = K' = S$ is a fixed point to (11). As $S \in \mathcal{S}(\Psi_0, \Psi_1)$, $S = \Pi^{S_{0,1}} = M^{S_{0,1}}$, see the review at the end of Section 3. Consequently, $\Psi_0 M_{|0}^{S_{0,1}} = \Psi_0 S_{|0} = S$ and $\Psi_1 M_{|1}^{S_{0,1}} = \Psi_1 S_{|1} = S$. In this case, the SB-optimal drifts $\mu_s$ and $\upsilon_s$ of $(S, \overleftarrow{S})$ respectively replace $\mu_m^{K'_{0,1}}$ and $\upsilon_m^{H'_{0,1}}$ in $(H^{K'_{0,1}}, \overleftarrow{K}^{H'_{0,1}})$. Under the additional assumption that $H' = K'$, or equivalently that $(H^{K'_{0,1}}, \overleftarrow{K}^{H'_{0,1}})$ are the time reversal of each other, this fixed point is unique. Let $G = H' = K'$, we have $G = \Psi_0 M_{|0}^{G_{0,1}} = G_0 M_{|0}^{G_{0,1}} = M_0^{G_{0,1}} M_{|0}^{G_{0,1}} = M^{G_{0,1}}$ and $G_0 = \Psi_0, G_1 = \Psi_1$, thus $G = \mathcal{S}(\Psi_0, \Psi_1) = S$. We have shown the following:

**Lemma 1** (Fixed points of (11)). *Under suitable conditions (Léonard, 2014a), the updates $(H', K') \overset{(11)}{\mapsto} (H^{K'_{0,1}}, K^{H'_{0,1}})$, parametrized by diffusion process distributions, admit $H' = K' = S$ as fixed point. If $H' = K'$, this fixed point is unique.*

When $\mu_m^{K'_{0,1}} = \mu_s$ and $\upsilon_m^{H'_{0,1}} = \upsilon_s$, (11) has reached an equilibrium. The updates $(H', K') \overset{(11)}{\mapsto} (H^{K'_{0,1}}, K^{H'_{0,1}})$ are realized through the computation of the drifts $\mu_m^{K'_{0,1}}$ and $\upsilon_m^{H'_{0,1}}$, i.e. by minimizing the losses (9, 10), where $Q_{0,1}$ is respectively equal to $K'_{0,1}$ and $H'_{0,1}$. Our proposal, BM$^2$, follows from replacing the complete minimization of (9, 10) with partial and stochastic minimization of (9, 10) through stochastic gradient descent. More precisely, consider the forward and backward SDEs with distributions $F(\theta)$ and $B(\theta)$:

$$X_0 \sim \Psi_0, \quad dX_t = \mu_f(X_t, t, \theta)dt + \sigma dW_t, \quad t \in [0,1], \tag{$F(\theta)$}$$

$$X_1 \sim \Psi_1, \quad dX_t = -v_b(X_t, t, \theta)dt + \sigma dW_t, \quad t \in [1, 0]. \qquad (\overleftarrow{B}(\theta))$$

$\mu_f(X_t, t, \theta)$ and $v_b(X_t, t, \theta)$ are drift functions to be learned, which are implemented through a neural network with parameters $\theta$. Let $\theta'$ represent the values of $\theta$ at a given step during training, and define the losses

$$\begin{aligned}
\mathbb{L}_f(\theta; \theta') &\coloneqq \mathop{\mathbb{E}}_{\Pi^{B_{0,1}(\theta')}} \left[ \frac{1}{2} \int_0^1 \|\mu_{01}(X_t, t, X_1) - \mu_f(X_t, t, \theta)\|^2 dt \right], \\
\mathbb{L}_b(\theta; \theta') &\coloneqq \mathop{\mathbb{E}}_{\Pi^{F_{0,1}(\theta')}} \left[ \frac{1}{2} \int_0^1 \|v_{01}(X_t, t, X_1) - v_b(X_t, t, \theta)\|^2 dt \right], \\
\mathbb{L}(\theta; \theta') &\coloneqq \mathbb{L}_f(\theta; \theta') + \mathbb{L}_b(\theta; \theta').
\end{aligned} \qquad (12)$$

At each optimization step, $\mathrm{BM}^2$ attempts to minimize $\mathbb{L}(\theta; \theta')$ in $\theta$ via a step of stochastic gradient descent, starting from $\theta = \theta'$ and keeping $\theta'$ fixed, resulting in $\theta''$. The subsequent optimization step employs $\theta' \leftarrow \theta''$. The complete training objective is presented in Algorithm 1, where sg() refers to the stop-gradient operator — $\mathbb{L}(\theta; \theta')$ is minimized in the first arguments only — and discretize() represents a generic SDE discretization scheme. For completeness, we outline the standard SGD training loop in Algorithm 2, where sgdstep() refers to an update step via a generic gradient descent optimizer.

It should be noted that merely performing coupled drift matching of $F$ and $B$, wherein $F$ learns the drift consistent with paths from $B$ and vice versa, does not yield the Schrödinger bridge as a fixed point (Bortoli et al., 2021). The introduction of the mixing process $\Pi$ is crucial in ensuring this property. Moreover, $\mathbb{L}(\theta; \theta')$ must be minimized only with respect to its first argument: the application of the stop-gradient operator sg() is not an efficiency consideration but a necessary component.

---

**Algorithm 1** $\mathrm{BM}^2$ — training loss computation

---

**outputs:** $\mathtt{l}(\theta)$: sampled loss value
**inputs:** $\theta$: current parameters

1: **function** loss($\theta$)
2:   $\mathtt{f}_0 \sim \Psi_0$              ▷ Marginal sampling
3:   $\mathtt{f}_{\Delta t}, \ldots, \mathtt{f}_1 | \mathtt{f}_0 \sim \mathrm{sg}(\mathrm{discretize}(\mathtt{f}_0, \Delta t, \mu_f(\cdot, \cdot, \theta)))$   ▷ Discretization of $(F(\theta))$
4:   $\mathtt{b}_1 \sim \Psi_1$              ▷ Marginal sampling
5:   $\mathtt{b}_{1-\Delta t}, \ldots, \mathtt{b}_0 | \mathtt{b}_1 \sim \mathrm{sg}(\mathrm{discretize}(\mathtt{b}_1, \Delta t, v_b(\cdot, \cdot, \theta)))$   ▷ Discretization of $(\overleftarrow{B}(\theta))$
6:   $\mathtt{t} \sim \mathcal{U}(0, 1)$             ▷ Time sampling
7:   $\pi\mathtt{f}_\mathtt{t} \sim R_{\mathtt{t}|0,1}(\cdot | \mathtt{f}_0, \mathtt{f}_1)$        ▷ Bridge sampling (5)
8:   $\pi\mathtt{b}_\mathtt{t} \sim R_{\mathtt{t}|0,1}(\cdot | \mathtt{b}_0, \mathtt{b}_1)$        ▷ Bridge sampling (5)
9:   $\mathtt{l}_\mathtt{f}(\theta) \leftarrow \frac{1}{2}\|\mu_{01}(\pi\mathtt{b}_\mathtt{t}, \mathtt{t}, \mathtt{b}_1) - \mu_f(\pi\mathtt{b}_\mathtt{t}, \mathtt{t}, \theta)\|^2$   ▷ BM based on $B_{0,1}$ (6, 9)
10:   $\mathtt{l}_\mathtt{b}(\theta) \leftarrow \frac{1}{2}\|v_{01}(\pi\mathtt{f}_\mathtt{t}, \mathtt{t}, \mathtt{f}_0) - v_b(\pi\mathtt{f}_\mathtt{t}, \mathtt{t}, \theta)\|^2$   ▷ BM based on $F_{0,1}$ (7, 10)
11:   $\mathtt{l}(\theta) \leftarrow \mathtt{l}_\mathtt{f}(\theta) + \mathtt{l}_\mathtt{b}(\theta)$
12:   **return** $\mathtt{l}(\theta)$

---

**Algorithm 2** $\mathrm{BM}^2$ — training loop

---

**outputs:** $\theta^*$: trained parameters
**inputs:** $\theta^\circ$: initial parameters

1: **function** train($\theta^\circ$)
2:   $\theta \leftarrow \theta^\circ$
3:   **while** not converged **do**
4:    $\mathtt{l}(\theta) \leftarrow \mathrm{loss}(\theta)$         ▷ Sample loss with Algorithm 1
5:    $\theta \leftarrow \mathrm{sgdstep}(\theta, \nabla_\theta \mathtt{l}(\theta))$     ▷ Perform SGD step
6:   **return** $\theta$

---

## 4.1 Implementation Aspects

The following aspects are not presented in Listing 1, but impacts the performance of $\mathrm{BM}^2$.

**Path Caching**: as in Bortoli et al. (2021); Shi et al. (2023), to enhance efficiency, we cache the initial and terminal endpoints of the paths sampled in lines 3 and 5 of Algorithm 1, and periodically refresh the cache during training. Notably, it is unnecessary to cache entire paths; only the endpoints are required for bridge sampling, which is advantageous from a memory perspective. Bridge sampling offers the additional benefit of increased sample diversity: for cached (fixed) endpoints, the samples corresponding to lines 7 and 8 differ at each step.

**Model**: we utilize a single neural network to parametrize both $\mu_f(x, t, \theta)$ and $v_b(x, t, \theta)$. As the training process is not iterative, it is unnecessary to introduce multiple neural networks (or parameters), one for each iteration.

**Sampling EMA**: as in Ho et al. (2020); Song et al. (2021), to improve the stability of training we apply the Exponential Moving Averaging (EMA) to the parameters employed in path sampling in lines 3 and 5 of Algorithm 1.

**Loss Singularities**: the losses of lines 9 and 10 of Algorithm 1 diverge for $t \to 1$ and $t \to 0$ respectively. Singularities of these kind are common to scalable losses for generative diffusion models. In our numerical experiments we simply restrict sampling of $t$ to $\mathcal{U}(\epsilon, 1 - \epsilon)$ for a small $\epsilon > 0$. More sophisticated alternatives involve either employing the dynamics of Appendix A for an appropriate scheduling $\beta_t$, or learning terminal-value predictors in place of drift terms, recovering the latter through (6, 7).

**Two-Stage Training**: a significant challenge in early training is the simulation-inference mismatch. To achieve reliable results, the drift functions $\mu_f(x_t, t, \theta)$ and $v_b(x_t, t, \theta)$ must be accurately learned in regions where the corresponding SDEs $(F(\theta))$ and $(\overleftarrow{B}(\theta))$ will be simulated. However, the processes $F(\theta)$ and $B(\theta)$ typically differ at initialization. Because the drift of $(F(\theta))$ is inferred from samples of $(\overleftarrow{B}(\theta))$ (and vice versa), the approximation quality can be poor; see Peluchetti (2023, Sections 2.3 and 6.2) for a detailed analysis of this issue within the DIPF framework. To address this challenge, the BM transport based on the independent coupling $\Psi_0 \otimes \Psi_1$ can be learned in both directions in a first stage. By construction, the BM transport circumvents the simulation-inference mismatch. Moreover, at convergence, the processes $F(\theta)$ and $B(\theta)$ are the same. Subsequently, in the second stage, the $\mathrm{BM}^2$ transport can be learned employing the first-stage solution as initialization.

**Forward-Backward Consistency**: the mutual time reversal relationship between $(F(\theta))$ and $(\overleftarrow{B}(\theta))$, i.e., the equivalence of $F(\theta)$ and $B(\theta)$, can be encouraged leveraging the diffusion time-reversal result of Anderson (1982), yielding the additional consistency loss term:

$$\mathcal{L}_{f,b}(\theta; \theta') := \mathop{\mathbb{E}}_{\frac{1}{2}(\Pi^{F_{0,1}}(\theta') + \Pi^{B_{0,1}}(\theta'))} \left[ \frac{1}{2} \int_0^1 \|\mu_f(X_t, t, \theta) + v_b(X_t, t, \theta) - \gamma_{01}(X_t, t, X_0, X_1)\|^2 dt \right], \quad (13)$$

where $\theta'$ denotes an independent copy of $\theta$ (implemented via the stop-gradient operator), and $\gamma_{01}(x_t, t, x_0, x_1)$ is defined in (8). Indeed, for a mixture process $\Pi^{C_{0,1}} = C_{0,1} R_{|0,1}$, the following relationship holds:

$$\sigma^2 \nabla \log \pi_t^{C_{0,1}}(x_t) = \mathop{\mathbb{E}}_{\Pi^{C_{0,1}}} [\gamma_{01}(X_t, t, X_0, X_1) | X_t = x_t].$$

Although (13) shares similarities with the consistency loss proposed by Shi et al. (2023, page 33), (13) utilizes the dynamic mixture $\frac{1}{2}(\Pi^{F(\theta')_{0,1}} + \Pi^{B(\theta')_{0,1}}) = \Pi^{\frac{1}{2}(F(\theta')_{0,1} + B(\theta')_{0,1})}$. When a single neural network is employed to parametrize both $\mu_f(x_t, t, \theta)$ and $v_b(x_t, t, \theta)$, estimating (13) in addition to (12) results in a negligible computational overhead per SGD step.

## 4.2 Convergence Properties

At each training step, $\mathrm{BM}^2$ performs a partial and stochastic minimization of the loss $\mathbb{L}(\theta; \theta')$ from (12) with respect to $\theta$, where $\mathbb{L}(\theta; \theta')$ is defined by an expectation over a distribution dependent on $\theta'$, yielding $\theta''$. Subsequently, $\theta'$ is updated to match $\theta''$, and the process advances to the next training step. The alternation between expectation and maximization steps bears resemblance to the classical Expectation-Maximization (EM) algorithm (Dempster et al., 1977).

### 4.2.1 Complete Minimization

We start by establishing in Theorem 1 that the version of $BM^2$ where $\mathbb{L}(\theta; \theta')$ is fully minimized at each training step recovers the I-BM and DIPF iterations for two specific initialization choices of $(F(\theta), \overleftarrow{B}(\theta))$. The prior convergence results of Bortoli et al. (2021); Shi et al. (2023); Peluchetti (2023) (see the review of Section 3.1) toward $S$ thus apply.

To facilitate the presentation of the convergence results in this section, we introduce, with a slight abuse of notation, the following functional versions of the losses (12):

$$
\begin{aligned}
\mathbb{L}_f(\mu_f; v_b') &:= \mathop{\mathbb{E}}_{\Pi^{B'_{0,1}}} \left[ \frac{1}{2} \int_0^1 \|\mu_{01}(X_t, t, X_1) - \mu_f(X_t, t)\|^2 dt \right], \\
\mathbb{L}_b(v_b; \mu_f') &:= \mathop{\mathbb{E}}_{\Pi^{F'_{0,1}}} \left[ \frac{1}{2} \int_0^1 \|v_{01}(X_t, t, X_1) - v_b(X_t, t)\|^2 dt \right], \\
\mathbb{L}(\mu_f, v_b; \mu_f', v_b') &:= \mathbb{L}_f(\mu_f; v_b') + \mathbb{L}_b(v_b; \mu_f').
\end{aligned}
\tag{14}
$$

In (14) we identify $\mu_f, v_b$ with $F, B$, and $\mu_f', v_b'$ with $F', B'$ (the remaining quantities defining $F, B, F', B'$ are fixed). We will use $\mathbb{L}_f(\mu_f; v_b'), \mathbb{L}_b(v_b; \mu_f')$ and $\mathbb{L}_f(\theta; \theta'), \mathbb{L}_b(\theta; \theta')$ interchangeably. We are now ready to state our first convergence result.

**Theorem 1** (Complete $BM^2$ Iterations). *Consider the SDEs $(F(\theta), \overleftarrow{B}(\theta))$, with initial drifts $\mu_f^{(0)}, v_b^{(0)}$ and corresponding distributions $F^{(0)}, B^{(0)}$. For each $i \geq 1$, let $(\mu_f^{(i)}, v_b^{(i)}) = \arg\min_{(\mu,v)} \mathbb{L}(\mu, v; \mu_f^{(i-1)}, v_b^{(i-1)})$, resulting in the distribution iterates $F^{(i)}, B^{(i)}$. We distinguish two cases:*

(i) *$\mu_f^{(0)} = v_b^{(0)} = 0$: both the iterates $F^{(0)}, B^{(1)}, F^{(2)}, \ldots$ and the iterates $B^{(0)}, F^{(1)}, B^{(2)}, \ldots$ are equivalent to the DIPF iterates, started respectively from the forward and from the backward time direction;*

(ii) *$\mu_f^{(0)} = \mu_m^{C_{0,1}}, v_b^{(0)} = v_m^{C_{0,1}}$ for some $C \in \mathcal{C}(\Psi_0, \Psi_1)$: $F^{(i)} = B^{(i)} = I^{(i)}$ for each $i \geq 0$ where $I^{(i)}$ are the I-BM iterates.*

### 4.2.2 Partial Minimization

In the EM algorithm it suffices to perform partial maximization steps. A partial result for the setting where $\mathbb{L}(\theta; \theta')$ is partially minimized with respect to $\theta$ at each step is stated in Theorem 2, which is based on Lemma 1 and Lemma 2.

**Lemma 2** (Loss Interpretation). *It holds that*

$$
\begin{aligned}
\mathbb{KL}(B_0 \parallel \Psi_0) + \mathbb{L}_f(\mu_f; v_b) &= \mathbb{KL}(\Pi^{B_{0,1}} \parallel F) + C_1(B) = \mathbb{KL}(M^{B_{0,1}} \parallel F) + C_2(B), \\
\mathbb{KL}(F_1 \parallel \Psi_1) + \mathbb{L}_b(v_b; \mu_f) &= \mathbb{KL}(\Pi^{F_{0,1}} \parallel B) + D_1(F) = \mathbb{KL}(M^{F_{0,1}} \parallel B) + D_2(F),
\end{aligned}
\tag{15}
$$

*for $C_1(B), C_2(B)$ independent of $F$, $D_1(F), D_2(F)$ independent of $B$, with $0 \leq C_1(B) \leq C_2(B)$ and $0 \leq D_1(F) \leq D_2(F)$.*

The losses $\mathbb{L}_f(\mu_f; v_b)$ and $\mathbb{L}_b(v_b; \mu_f)$ are easily amenable to optimization in their first arguments, as seen in Algorithm 1. Lemma 2 relates these losses to more interpretable KL divergences between distributions. By (15), a decrease of $\mathbb{L}_f(\mu_f; v_b)$ due to a change in $\mu_f$ corresponds to equivalent decreases of $\mathbb{KL}(M^{B_{0,1}} \parallel F)$ for a fixed $v_b$, or $B$. Thus, partial minimization of $\mathbb{L}_f(\mu_f; v_b)$ brings $F$ closer to $M^{B_{0,1}}$, the BM transport based on $B_{0,1}$, and the result of a complete minimization step, by means of reverse KL minimization. Symmetric considerations apply to $\mathbb{L}_b(v_b; \mu_f)$ as function of its first argument. Putting this result and Lemma 1 together yields Theorem 2.

**Theorem 2** (Partial $BM^2$ Iterations). *At each optimization step, decreases of $\mathbb{L}_f(\theta; \theta')$ and $\mathbb{L}_b(\theta; \theta')$ in $\theta$ correspond to equivalent decreases of $\mathbb{KL}(M^{B_{0,1}(\theta')} \parallel F(\theta))$ and $\mathbb{KL}(M^{F_{0,1}(\theta')} \parallel B(\theta))$. If the losses $\mathbb{L}_f(\theta; \theta')$ and $\mathbb{L}_b(\theta; \theta')$ cannot be decreased in $\theta$, i.e., at optimality, and if $F(\theta) = B(\theta)$, then $F(\theta) = B(\theta) = S$.*

### 4.2.3 Infinitesimal Minimization

We conclude our theoretical investigation by relating our proposal to the work of Karimi et al. (2023), which introduces a continuous variant of the IPF procedure. In IPF, the two target marginal distributions are replaced sequentially, one at a time. Each step corresponds to solving a static Schrödinger *half-bridge* problem (Léonard, 2014a), where in (2), $\mathcal{C}(\Psi_0, \Psi_1)$ is replaced by either $\mathcal{C}(\Psi_0, \cdot)$ or $\mathcal{C}(\cdot, \Psi_1)$. The approach proposed by Karimi et al. (2023) retains either the even or odd steps of the IPF scheme while substituting the alternate steps with partial minimizations of the corresponding half-bridge problems. In the limit of infinitesimally small improvements, this yields a dynamical system for the evolution of the iterates over continuous algorithmic time.

We demonstrate that a similar result can be obtained for a modified version of $\mathrm{BM}^2$, where forward KL divergences are minimized instead of reverse KL divergences. The resulting dynamical system is a symmetrized version of the one obtained by Karimi et al. (2023). Let $F'$, $B'$ represent the current state in the optimization process. We consider a partial minimization of $\mathbb{KL}(F \parallel M^{B'_{0,1}})$, instead of $\mathbb{KL}(M^{B'_{0,1}} \parallel F)$, in $F$ and a partial minimization of $\mathbb{KL}(B \parallel M^{F'_{0,1}})$, instead of $\mathbb{KL}(M^{F'_{0,1}} \parallel B)$, in $B$. As in Karimi et al. (2023), partial minimization is formulated as

$$
\begin{aligned}
F^{(\lambda)} &:= \underset{F \in \mathcal{M}(\Psi_0, \cdot)}{\arg\min} \ \lambda\mathbb{KL}(F \parallel M^{B'_{0,1}}) + (1-\lambda)\mathbb{KL}(F \parallel F'), \\
B^{(\lambda)} &:= \underset{B \in \mathcal{M}(\cdot, \Psi_1)}{\arg\min} \ \lambda\mathbb{KL}(B \parallel M^{F'_{0,1}}) + (1-\lambda)\mathbb{KL}(B \parallel B'),
\end{aligned}
\tag{16}
$$

where $\lambda \in [0,1]$ controls the extent of the minimization. We begin by establishing two stability results: the updates $(F', B') \overset{(16)}{\to} (F^{(\lambda)}, B^{(\lambda)})$ preserve both $\mathcal{R}$ and $\mathcal{S}$.

**Lemma 3** ($\mathcal{R}$-stability of $F^{(\lambda)}, B^{(\lambda)}$). *If $F', B' \in \mathcal{R}$, then $F^{(\lambda)}, B^{(\lambda)} \in \mathcal{R}$ for each $\lambda \in [0,1]$.*

**Lemma 4** ($\mathcal{S}$-stability of $F^{(\lambda)}, B^{(\lambda)}$). *If $F', B' \in \mathcal{S}$, then $F^{(\lambda)}, B^{(\lambda)} \in \mathcal{S}$ for each $\lambda \in [0,1]$.*

Provided that the initial values $F', B' \in \mathcal{S}$, Lemma 4 establishes that the iterates defined by the updates $(F', B') \overset{(16)}{\to} (F^{(\lambda)}, B^{(\lambda)})$ always remain in $\mathcal{S}$. It is straightforward to ensure that $F', B' \in \mathcal{S}$ at initialization by setting the corresponding drifts to zero: $\mu'_f, v'_b = 0$, which we will assume henceforth. As $M^{B'_{0,1}} = B'$ and $M^{F'_{0,1}} = F'$, (16) can be reformulated in simpler terms:

$$
\begin{aligned}
F^{(\lambda)} &:= \underset{F \in \mathcal{M}(\Psi_0, \cdot)}{\arg\min} \ \lambda\mathbb{KL}(F \parallel B') + (1-\lambda)\mathbb{KL}(F \parallel F'), \\
B^{(\lambda)} &:= \underset{B \in \mathcal{M}(\cdot, \Psi_1)}{\arg\min} \ \lambda\mathbb{KL}(B \parallel F') + (1-\lambda)\mathbb{KL}(B \parallel B').
\end{aligned}
\tag{17}
$$

By Lemma 3, it suffices to solve (17) in the static setting,

$$
\begin{aligned}
F^{(\lambda)}_{0,1} &:= \underset{F_{0,1} \in \mathcal{C}(\Psi_0, \cdot)}{\arg\min} \ \lambda\mathbb{KL}(F_{0,1} \parallel B'_{0,1}) + (1-\lambda)\mathbb{KL}(F_{0,1} \parallel F'_{0,1}), \\
B^{(\lambda)}_{0,1} &:= \underset{B_{0,1} \in \mathcal{C}(\cdot, \Psi_1)}{\arg\min} \ \lambda\mathbb{KL}(B_{0,1} \parallel F'_{0,1}) + (1-\lambda)\mathbb{KL}(B_{0,1} \parallel B'_{0,1}).
\end{aligned}
\tag{18}
$$

The dynamic solutions are then recovered by $F^{(\lambda)}_{|0,1} = B^{(\lambda)}_{|0,1} = R_{|0,1}$.

We assume that $F'_{0,1}, B'_{0,1}, \Psi_0, \Psi_1$ admit densities. By calculus of variations, the solution to (18) is given by $f^{(\lambda)}_{0,1}(x_0, x_1) = \psi_0(x_0) f^{(\lambda)}_{1|0}(x_1|x_0)$, and $b^{(\lambda)}_{0,1}(x_0, x_1) = b^{(\lambda)}_{0|1}(x_0|x_1)\psi_1(x_1)$, where $f^{(\lambda)}_{1|0}(x_1|x_0) \propto b'_{1|0}(x_1|x_0)^{\lambda} f'_{1|0}(x_1|x_0)^{1-\lambda}$ and $b^{(\lambda)}_{0|1}(x_0|x_1) \propto f'_{0|1}(x_0|x_1)^{\lambda} b'_{0|1}(x_0|x_1)^{1-\lambda}$. The IPF iterations are recovered when $\lambda = 1$. Instead, taking the limit $\lambda \to 0$ and applying Bayes theorem, we obtain evolution of

$\log f_{1|0}^{(l)}(x_1|x_0)$ and $\log b_{0|1}^{(l)}(x_0|x_1)$ as a function of algorithmic time $l \in [0, \infty)$ through the dynamical system

$$
\begin{aligned}
\frac{d \log f_{1|0}^{(l)}(x_1|x_0)}{dl} &= -\log \frac{f_{1|0}^{(l)}(x_1|x_0)}{b_{0|1}^{(l)}(x_0|x_1)\psi_1(x_1)} + \overline{\mathbb{KL}}(f_{1|0}^{(l)}(x_1|x_0) \parallel b_{0|1}^{(l)}(x_0|x_1)\psi_1(x_1)), \quad l \in [0,\infty), \\
\frac{d \log b_{0|1}^{(l)}(x_0|x_1)}{dl} &= -\log \frac{b_{0|1}^{(l)}(x_0|x_1)}{f_{1|0}^{(l)}(x_1|x_0)\psi_0(x_0)} + \overline{\mathbb{KL}}(b_{0|1}^{(l)}(x_0|x_1) \parallel f_{1|0}^{(l)}(x_1|x_0)\psi_0(x_0)), \quad l \in [0,\infty).
\end{aligned}
\tag{19}
$$

In (19), $\overline{\mathbb{KL}}(\cdot \parallel \cdot)$ denotes the generalized KL divergence between unnormalized densities, as is the case here for the second arguments, and the initial conditions $f_{1|0}^{(0)}(x_1|x_0)$ and $b_{0|1}^{(0)}(x_0|x_1)$ are determined by $(F(\theta), \overleftarrow{B}(\theta))$ with null drift terms. (19) can be contrasted with Karimi et al. (2023, Equation (13)). In Appendix C.1 we report a simple numerical application of (19) to the Gaussian setting, which recovers $S$.

## 5 Numerical Experiments

To evaluate the performance of $\text{BM}^2$ on EOT problems, we utilize the benchmark developed by Gushchin et al. (2023). For the reference process ($R$), this benchmark provides pairs of target distributions $\Psi_0, \Psi_1$ with analytical EOT solution $S_{0,1}$ and analytical SB-optimal drift function $\mu_s$. We focus on the mixtures benchmark, which consist of a centered Gaussian distribution as $S_0 = \Psi_0$ and a mixture of 5 Gaussian distributions for $S_{1|0}$. $S_1 = \Psi_1$ is not a mixture of Gaussian distributions, but has 5 distinct modes. The benchmark is constructed for dimensions $d \in \{2, 16, 64, 128\}$ and entropic regularization parameters $\varepsilon \in \{0.1, 1, 10\}$.

For each fully trained method, characterized by a stochastic process distribution $P$ and forward drift function $\mu_p$, we assess performance using two evaluation metrics:

- $\mathbb{KL}(S \parallel P)$ where, by Girsanov theorem (Øksendal, 2013),

$$
\mathbb{KL}(S \parallel P) = \mathbb{E}_S \left[ \frac{1}{2\sigma^2} \int_0^1 \|\mu_s(X_t, t) - \mu_p(X_t, t)\|^2 dt \right];
\tag{20}
$$

- $\text{cBW}_2^2\text{-UVP}(S_{0,1}, P_{0,1})$, where

$$
\text{cBW}_2^2\text{-UVP}(S_{0,1}, P_{0,1}) := \frac{100}{\frac{1}{2}\mathbb{V}_S[X_1]} \int \text{BW}_2^2(S_{1|0}(X_1|X_0), P_{1|0}(X_1|X_0)) S_0(dX_0),
\tag{21}
$$

$\text{BW}_2^2(\cdot, \cdot)$ is the squared Bures-Wasserstein distance, i.e. the squared Wasserstein-2 distance between (assumed) multivariate Gaussian distributions (Dowson & Landau, 1982), and $\mathbb{V}_S[X_1]$ is the variance of $X_1 \sim S_1$.

We focus on the divergence $\mathbb{KL}(S \parallel P)$, rather than $\mathbb{KL}(P \parallel S)$, as a low $\mathbb{KL}(S \parallel P)$ more accurately indicates that $P$ approximates $S$ effectively across the entire support of $S$. The data-processing inequality implies that $\mathbb{KL}(S_{0,1} \parallel P_{0,1}) \leq \mathbb{KL}(S \parallel P)$. The $\text{cBW}_2^2\text{-UVP}(\cdot, \cdot)$ metric, introduced by Gushchin et al. (2023), is a normalized and conditional extension of the standard $\text{BW}_2^2(\cdot, \cdot)$ distance. Results for evaluation metrics (20) and (21) are summarized in Table 1 and Table 2, respectively.

In our benchmarking, we compare $\text{BM}^2$ against the I-BM and DIPF methods (Section 3.1). Each experiment is repeated five times, including both model training and metric evaluation, to obtain uncertainty quantification. We use $1,000$ Monte Carlo samples to estimate (20, 21). For simplicity, we employ the Euler–Maruyama scheme (Kloeden & Platen, 1992) with 200 discretization steps ($\Delta t = 0.005$) in all path sampling procedures. Each method undergoes $50,000$ SGD training steps with a batch size of $1,000$, settings similar to those used by Gushchin et al. (2023), enabling qualitative comparison of our results with theirs. We use the AdamW optimizer with a learning rate of $10^{-4}$ and hyperparameters: $\beta = (0.9, 0.999), \epsilon = 10^{-8}, wd = 0.01$, where $wd$ denotes weight decay. Time is sampled as $t \sim \mathcal{U}(\epsilon, 1 - \epsilon)$ for $\epsilon = 0.0025$.

| | $\varepsilon=0.1$ | | | | $\varepsilon=1$ | | | | $\varepsilon=10$ | | | |
|---|---|---|---|---|---|---|---|---|---|---|---|---|
| Method | $d=2$ | $d=16$ | $d=64$ | $d=128$ | $d=2$ | $d=16$ | $d=64$ | $d=128$ | $d=2$ | $d=16$ | $d=64$ | $d=128$ |
| $\text{BM}^2$ | **0.01** 
0.01 | **0.20** 
0.02 | **1.03** 
0.07 | **3.06** 
0.16 | **0.01** 
0.00 | **0.11** 
0.00 | **1.43** 
0.03 | 8.29 
0.36 | **0.11** 
0.01 | **2.25** 
0.04 | **13.13** 
0.13 | **40.46** 
0.49 |
| $\text{BM}^2_\sigma$ | 0.43 
0.09 | 3.76 
0.46 | 39.55 
1.96 | 127.2 
1.4 | 0.04 
0.01 | 0.43 
0.03 | 5.36 
0.35 | 18.66 
0.73 | 0.15 
0.00 | 2.64 
0.05 | 13.78 
0.24 | 43.42 
1.43 |
| I-BM | 0.03 
0.01 | 0.20 
0.02 | 1.24 
0.04 | 5.70 
0.42 | 0.01 
0.00 | 0.16 
0.01 | 1.94 
0.04 | **7.79** 
0.07 | 0.16 
0.00 | 4.09 
0.03 | 17.17 
0.21 | 49.17 
0.55 |
| DIPF | 0.59 
0.14 | 2.39 
0.05 | 7.93 
1.23 | 34.77 
0.82 | 0.23 
0.06 | 1.21 
0.18 | 13.13 
0.79 | 36.51 
1.05 | 0.81 
0.06 | 28.25 
2.12 | 113.8 
7.2 | 345.8 
8.1 |

Table 1: Monte Carlo estimate of $\mathbb{KL}(S \parallel P)$ as function of $\varepsilon$ and $d$, standard deviation in gray.

For $\text{BM}^2$, we employ a single feedforward neural network with 3 layers of width 768 and ReLU activation, resulting in approximately 1 million parameters. We initialize the neural network parameters such that the resulting initial forward and backward drift functions evaluate to zero everywhere, a choice that has proven effective in our experiments. As mentioned in Section 4.1, we implement path caching and an exponential moving average for parameters used in path sampling. The cache contains $5,000$ initial-terminal values from both $(F(\theta))$ and $(\overleftarrow{B}(\theta))$, refreshed every 200 training steps. We omit reporting the results for both the two-stage training procedure and the consistency loss described in Section 4.1, as neither approach demonstrated consistent performance improvements across the considered benchmark.

For I-BM and DIPF, each outer loop iteration comprises $5,000$ SGD steps, totaling 10 outer loop (algorithmic) iterations. Following best practices (Bortoli et al., 2021; Shi et al., 2023), we alternate time directions over iterations for both algorithms. Each method employs two separate neural networks for forward and backward time directions, maintaining a total parameter count close to 1 million, matching $\text{BM}^2$'s model size. As with $\text{BM}^2$, we implement path caching (for DIPF, entire discretized paths are cached) and EMA for sampling, with an EMA decay rate of 0.99.

We also consider $\text{BM}^2_\sigma$, a variant of $\text{BM}^2$ that learns Schrödinger bridges for $\Psi_0, \Psi_1$ across multiple $\sigma$ values. This amortized version leverages $\text{BM}^2$'s non-iterative nature. At each optimization step, $\sigma$ is sampled from $\mathcal{U}(0.1, 4)$ and utilized in discretizing SDEs $(F(\theta), \overleftarrow{B}(\theta))$ (lines 3 and 5 of Algorithm 1) and in bridge sampling (lines 7 and 8 of Algorithm 1). The neural network implementing drift functions $\mu_f(x,t,\theta)$ and $v_b(x,t,\theta)$ is modified to accept $\sigma$ as an additional input, resulting in conditional drift functions $\mu_f(x,t,\theta,\sigma)$ and $v_b(x,t,\theta,\sigma)$. Path caching is adjusted to store $\sigma$ values corresponding to cached paths.

In Table 2, we additionally include three baselines. EOT: sampling from the EOT solution, accounting for the bias due to Monte Carlo estimation. SB(discr): sampling from the SB solution via the SB-optimal drift $\mu_s$, additionally accounting for Euler–Maruyama scheme discretization error. $\Psi_0 \otimes \Psi_1$: sampling from the independent coupling. Results for additional discretization intervals are reported in Appendix C.2, and we illustrate in Figure 1 the evolution of (21) during training for a representative benchmark setting.

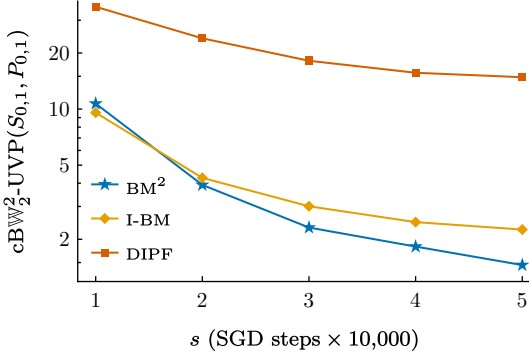

Figure 1: Evolution of the metric (21) for $d = 64$, $\varepsilon = 1$ over SGD steps for $\text{BM}^2$, I-BM and DIPF.

| Method | $\varepsilon=0.1$ | | | | $\varepsilon=1$ | | | | $\varepsilon=10$ | | | |
|---|---|---|---|---|---|---|---|---|---|---|---|---|
| | $d=2$ | $d=16$ | $d=64$ | $d=128$ | $d=2$ | $d=16$ | $d=64$ | $d=128$ | $d=2$ | $d=16$ | $d=64$ | $d=128$ |
| EOT | 0.02 
 0.00 | 0.05 
 0.00 | 0.34 
 0.00 | 0.91 
 0.00 | 0.09 
 0.00 | 0.17 
 0.00 | 0.43 
 0.00 | 1.14 
 0.00 | 0.12 
 0.00 | 0.18 
 0.00 | 0.23 
 0.00 | 0.38 
 0.00 |
| SB(discr.) | 0.04 
 0.00 | 0.07 
 0.00 | 0.35 
 0.00 | 0.92 
 0.00 | 0.10 
 0.00 | 0.17 
 0.00 | 0.45 
 0.00 | 1.18 
 0.00 | 0.12 
 0.00 | 1.15 
 0.00 | 5.38 
 0.01 | 10.48 
 0.01 |
| $\Psi_0 \otimes \Psi_1$ | 195.8 
 6.9 | 186.3 
 2.4 | 162.6 
 0.8 | 145.1 
 2.1 | 136.1 
 4.7 | 127.6 
 1.4 | 113.0 
 1.7 | 93.61 
 1.57 | 8.07 
 0.33 | 4.88 
 0.14 | 4.22 
 0.09 | 4.45 
 0.07 |
| $\text{BM}^2$ | **0.73** 
 0.40 | 4.64 
 0.58 | **6.84** 
 0.59 | **8.28** 
 0.62 | **0.14** 
 0.03 | **0.41** 
 0.04 | **1.72** 
 0.09 | 8.30 
 1.17 | **0.14** 
 0.01 | **2.30** 
 0.04 | **41.14** 
 1.99 | 264.4 
 7.0 |
| $\text{BM}^2_\sigma$ | 8.38 
 3.40 | 16.06 
 2.81 | 44.15 
 0.84 | 83.84 
 0.92 | 0.20 
 0.07 | 2.61 
 0.41 | 25.89 
 1.65 | 64.76 
 2.66 | 0.14 
 0.01 | 2.57 
 0.03 | 58.76 
 0.76 | 323.0 
 8.8 |
| I-BM | 1.07 
 0.50 | **4.25** 
 0.66 | 7.19 
 0.28 | 16.63 
 2.07 | 0.20 
 0.09 | 0.53 
 0.04 | 2.20 
 0.35 | **7.79** 
 0.79 | 0.14 
 0.02 | 5.21 
 0.11 | 135.8 
 1.3 | 578.7 
 9.8 |
| DIPF | 7.82 
 2.51 | 15.30 
 1.00 | 20.12 
 1.53 | 29.36 
 1.02 | 1.66 
 0.24 | 5.98 
 0.65 | 13.11 
 2.49 | 28.86 
 3.52 | 0.69 
 0.08 | 6.85 
 0.21 | 72.63 
 0.70 | **226.1** 
 1.1 |

Table 2: Monte Carlo estimate of $\text{cB}\mathbb{W}_2^2\text{-UVP}(S_{0,1}, P_{0,1})$ as function of $\varepsilon$ and $d$, standard deviation in gray.

We now discuss the results presented in Tables 1 and 2. $\text{BM}^2$ demonstrates superior overall performance across dimensions and entropic regularization settings in both metrics. I-BM also shows good performance, particularly in comparison to the DIPF procedure, which aligns with the findings of Shi et al. (2023).

As expected, the performance of all methods deteriorates as the number of dimensions increases. This is because the metric(20) scales linearly with the number of dimensions, assuming a constant error rate in estimating each component of the true drift $\mu_s$. Similar considerations apply to the metric (21).

While $\text{BM}^2_\sigma$ exhibits a performance gap compared to $\text{BM}^2$, it yields reasonable results in low-dimensional settings ($d = 2, 16$). This gap may be due to increased pressure on model capacity or the need to normalize loss levels across $\sigma$ values. All methods perform poorly in the high regularization setting ($\varepsilon = 10$), especially in high dimensions ($d = 64, 128$), which we include for completeness. It should be noted that, in such cases, sampling from the independent coupling (a trivial solution) is preferable to sampling from the SB-optimal SDE for the chosen discretization interval.

## 6 Related Works

Relevant works that, like $\text{BM}^2$, address the *dynamic* Schrödinger bridge problem (1) include:

**Schrödinger Bridge Flow**: in a concurrent work, De Bortoli et al. (2024) propose a non-iterative methodology called $\alpha$-DSBM (Diffusion Schrödinger Bridge Matching). $\alpha$-DSBM is optimally implemented through a forward-backward SDE approach, in which case $\alpha$-DSBM's formulation and training objective align with our proposal (12, $F(\theta)$, $\overleftarrow{B}(\theta)$), see De Bortoli et al. (2024, Equations (10) and (11)). The formulation of $\alpha$-DSBM begins by establishing a probability flow in the space of path probability measures, whose discretization yields $\alpha$-IMF (Iterative Markovian Fitting). Under mild conditions, two theoretical results are established: (i) the $\alpha$-IMF iterates converge to the SB solution, and (ii) the non-parametric updates of the functional loss (14), obtained through functional gradient descent with respect to the drift functions, recover the $\alpha$-IMF iterates. The practical implementation, $\alpha$-DSBM, thus replaces non-parametric drift functions with parametric neural network approximators, and employs standard stochastic gradient descent on the parametric loss (12). This theoretical framework provides strong convergence guarantees for $\alpha$-DSBM and paves the way to further theoretical developments. Furthermore, De Bortoli et al. (2024) demonstrate the method's effectiveness through extensive numerical experiments on high-dimensional computer vision problems, complementing our synthetic benchmark results.

**I-BM and DIPF**: closely related to $\text{BM}^2$ are the iterative, sample-based DIPF (Bortoli et al., 2021; Vargas et al., 2021) and I-BM (Shi et al., 2023; Peluchetti, 2023) procedures, which do not satisfy desiderata (i). Built on similar bridge matching principles, $\text{BM}^2$ can be viewed as a modification of I-BM that employs a single optimization loop, resulting in a simpler algorithm that we have empirically shown to be competitive.

**Forward-Backward SB SDE**: Chen et al. (2022) propose two training algorithms addressing the dynamic SB problem. Both approaches employ loss functions that require divergence computations (violating desiderata (iv)) and the use of two distinct neural networks. The first method is iterative, resembling DIPF (violating desiderata (i)), while the second method involves differentiating through entire discretized paths, resulting in high memory consumption (violating desiderata (iii)).

The subsequent works concentrate on solving the *static* Schrödinger bridge (2), or EOT (3), problem. Once this is achieved, solutions to the dynamic problem are trivially obtained through the standard decomposition $S = S_{0,1}R_{|0,1}$. Although these works differ in nature and objectives, we include them here due to their shared characteristic with BM$^2$: the non-iterative nature of the algorithm.

**Light SB**: in two notable works, Korotin et al. (2023) and Gushchin et al. (2024) propose non-iterative, sample-based EOT solvers for the Euclidean cost function, i.e., for the specific choice of reference dynamics ($R$). Korotin et al. (2023) introduces an approximation to (an adjusted version of) the Schrödinger potential for $\Psi_1$ via a mixture of Gaussian distributions, resulting in a mixture of Gaussian distributions approximation to $S_{1|0}$. Gushchin et al. (2024) builds upon this approximation and introduces an additional sample-based training objective that takes as input any coupling $C_{0,1} \in \mathcal{C}(\Psi_0, \Psi_1)$, whereas Korotin et al. (2023) requires the independent coupling $\Psi_0 \otimes \Psi_0$. While also non-iterative, the proposals of Korotin et al. (2023); Gushchin et al. (2024) differ from BM$^2$ in two key aspects: (a) they learn a solution in the static setting instead of the dynamic one, and (b) they employ mixture of Gaussian distributions approximations, rather than neural network approximators for the drift functions. Consequently, these methods may face challenges in scaling to modern generative ML applications. Light SB, in both variants, demonstrates strong performance in the benchmark presented in Section 5: the results of (Gushchin et al., 2024, Table 1) and (Korotin et al., 2023, Table 2) are directly comparable with the results of Table 2. However, it is worth noting that this benchmark is particularly well-suited for Light SB, as acknowledged by its authors, since each benchmark target $S_{0,1}$ is constructed such that $S_{1|0}$ is itself a mixture of 5 Gaussian distributions.

## 7 Conclusions

In this work we introduced Coupled Bridge Matching (BM$^2$), a novel approach for learning Schrödinger bridges from samples. BM$^2$ builds on the principles of Bridge Matching while addressing key limitations of existing iterative methods. Our approach offers several advantages, including a simple single-loop optimization procedure, exactness in the idealized setting, modest memory requirements, and a straightforward loss function. The numerical experiments demonstrate that BM$^2$ is competitive with and often outperforms existing iterative diffusion-based methods like I-BM and DIPF across various dimensions and entropic regularization settings.

On the theoretical front, there is substantial room for improvement. Firstly, while bearing some resemblance to the standard convergence result for the EM algorithm, Theorem 2 lacks a quantity analogous to the likelihood being maximized in the EM algorithm. It remains unclear whether decreases in $\mathbb{KL}(M^{B_{0,1}(\theta')} \| F(\theta))$ and $\mathbb{KL}(M^{F_{0,1}(\theta')} \| B(\theta))$ can be linked to decreases in $\mathbb{KL}(F(\theta) \| S)$ and $\mathbb{KL}(B(\theta) \| S)$. Secondly, the requirement that $F(\theta) = B(\theta)$, equivalently that $(F(\theta))$ and $(\overleftarrow{B}(\theta))$ are time-reversals of each other, appears unnecessary. Notably, all numerical simulations conducted do not explicitly enforce this condition, which emerges naturally during the training process. Thirdly, it would be valuable to study problem (16) where reverse KL divergences are partially minimized, aligning more closely with the BM$^2$ algorithm. In this scenario, Lemma 4 no longer holds, and it may be necessary to impose a corresponding additional constraint to maintain tractable analytical computations. The attractors of (19), and of a corresponding dynamical system arising from reverse KL minimization, can be investigated to assess further convergence properties of BM$^2$.

On the empirical front, the applications of BM$^2$ in contemporary generative machine learning tasks remain unexplored. Given the promising results from previous studies employing Bridge Matching, such as those by Liu et al. (2023) and Somnath et al. (2023), it is anticipated that BM$^2$ could be effectively applied to various domains, including image generation, audio synthesis, and molecular design. Future work could investigate the scalability and performance of BM$^2$ in these domains.

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

## A    Additional Dynamics

In this section we consider a simple extension to the dynamics of Section 2.2, and refer the reader to Peluchetti (2021; 2023) for the more general case. Here, we consider the case where the reference distribution $R$ is given by the solution to:

$$X_0 \sim \Psi_0, \quad dX_t = \sigma \sqrt{\beta_t} dW_t, \quad t \in [0,1], \tag{22}$$

with $\sigma \geq 0$, $\beta_t : [0,1] \to \mathbb{R}_{>0}$ strictly positive and continuous. With $b_{s:t} := \int_s^t \beta_u du$, $0 \leq s \leq t \leq 1$, $\beta_t$ is chosen such that $b_{0:1} = 1$, to disentangle the contribution of $\beta_t$ from the contribution of $\sigma$. Indeed, under these conditions, $\beta_t$ defines a time-warping: if $X_t$ is the solution to ($R$), then $X_{b_{0:t}}$ has the same distribution as the solution to (22). Consequently, the solutions to (2) and (3) are independent of $\beta_t$.

When employing (22), the definitions in Section 2.2 are replaced as follows:

$$R_{t|0}(\cdot | x_0) = \mathcal{N}(x_0, \sigma^2 b_{0:t}), \tag{23}$$

$$R_{t|0,1}(\cdot | x_0, x_1) = \mathcal{N}(x_0 b_{t:1} + x_1 b_{0:t}, \sigma^2 b_{0:t} b_{t:1}), \tag{24}$$

$$\mu_{01}(x_t, t, x_1) := \sigma^2 \beta_t \nabla_{x_t} \log r_{1|t}(x_1|x_t) = \frac{\beta_t}{b_{t:1}}(x_1 - x_t), \tag{25}$$

$$v_{01}(x_t, t, x_0) := \sigma^2 \beta_t \nabla_{x_t} \log r_{t|0}(x_t|x_0) = \frac{\beta_t}{b_{0:t}}(x_0 - x_t), \tag{26}$$

$$\gamma_{01}(x_t, t, x_0, x_1) := \sigma^2 \beta_t \nabla_{x_t} \log r_{t|01}(x_t|x_0, x_1) = \frac{\beta_t}{b_{0:t} b_{t:1}}(x_0 b_{t:1} + x_1 b_{0:t} - x_t). \tag{27}$$

## B    Proofs

**Theorem 1** (Complete BM$^2$ Iterations). *Consider the SDEs ($F(\theta), \overleftarrow{B}(\theta)$), with initial drifts $\mu_f^{(0)}, v_b^{(0)}$ and corresponding distributions $F^{(0)}, B^{(0)}$. For each $i \geq 1$, let $(\mu_f^{(i)}, v_b^{(i)}) = \arg\min_{(\mu,v)} \mathbb{L}(\mu, v; \mu_f^{(i-1)}, v_b^{(i-1)})$, resulting in the distribution iterates $F^{(i)}, B^{(i)}$. We distinguish two cases:*

*(i) $\mu_f^{(0)} = v_b^{(0)} = 0$: both the iterates $F^{(0)}, B^{(1)}, F^{(2)}, \ldots$ and the iterates $B^{(0)}, F^{(1)}, B^{(2)}, \ldots$ are equivalent to the DIPF iterates, started respectively from the forward and from the backward time direction;*

*(ii) $\mu_f^{(0)} = \mu_m^{C_{0,1}}, v_b^{(0)} = v_m^{C_{0,1}}$ for some $C \in \mathcal{C}(\Psi_0, \Psi_1)$: $F^{(i)} = B^{(i)} = I^{(i)}$ for each $i \geq 0$ where $I^{(i)}$ are the I-BM iterates.*

*Proof.* Define $Q$ associated with

$$X_1 \sim \Psi_1, \quad dX_t = \sigma dW_t, \quad t \in [1,0], \tag{$\overleftarrow{Q}$}$$

which is not the time reversal of ($R$), but $R_{|0,1} = Q_{|0,1}$.

Firstly, consider the case of initial null drifts: $\mu_f^{(0)} = v_b^{(0)} = 0$, corresponding to $F^{(0)} = \Psi_0 R_{|0} = \Psi_0 R_{1|0} R_{|0,1} = F_{0,1}^{(0)} R_{|0,1} \in \mathcal{S}$ and $B^{(0)} = \Psi_1 Q_{|1} = \Psi_1 Q_{0|1} R_{|0,1} = B_{0,1}^{(0)} R_{|0,1} \in \mathcal{S}$. As $B^{(0)} = \Pi^{B_{0,1}^{(0)}} = M^{B_{0,1}^{(0)}}$, we have $F^{(1)} = \Psi_0 M_{|0}^{B_{0,1}^{(0)}} = \Psi_0 B_{|0}^{(0)} = \Psi_0 B_{1|0}^{(0)} R_{|0,1} \in \mathcal{S}$. As $F^{(0)} = \Pi^{F_{0,1}^{(0)}} = M^{F_{0,1}^{(0)}}$, $B^{(1)} = \Psi_1 M_{|1}^{F_{0,1}^{(0)}} = \Psi_1 F_{|1}^{(0)} = \Psi_1 F_{0|1}^{(0)} R_{|0,1} \in \mathcal{S}$. By induction, $F^{(i)} = \Psi_0 B_{1|0}^{(i-1)} R_{|0,1} \in \mathcal{S}$ and $B^{(i)} = \Psi_1 F_{0|1}^{(i-1)} R_{|0,1} \in \mathcal{S}$, $i \geq 1$. We now construct two forward-backward sequences. For the sequence $F^{(0)}, B^{(1)}, F^{(2)}, \ldots$, we have $F_{0,1}^{(0)} = \Psi_0 R_{1|0}$, $B_{0,1}^{(1)} = \Psi_1 F_{0|1}^{(0)}$, $F_{0,1}^{(2)} = \Psi_0 B_{1|0}^{(1)}$, $\ldots$ which are the static IPF iterates: one marginal gets replaced at a time keeping the conditional distribution fixed. In the same way, for $B^{(0)}, F^{(1)}, B^{(2)}, \ldots$, we have $B_{0,1}^{(0)} = \Psi_1 Q_{0|1}$, $F_{0,1}^{(1)} = \Psi_0 B_{1|0}^{(0)}$, $B_{0,1}^{(2)} = \Psi_1 F_{0|1}^{(1)}$, $\ldots$ which are again the static IPF iterates (for the backward formulation of the dynamic SB problem, i.e. via $\overleftarrow{Q}_{|0}$ as reference measure instead of $R_{|0}$, and switched marginal distributions).

As each pair $F^{(i)}, B^{(i)}$ is of the form $F^{(i)} = F^{(i)}_{0,1} R_{|0,1}$, $B^{(i)} = B^{(i)}_{0,1} R_{|0,1}$, we also recover the dynamic DIPF iterates.

Secondly, consider $\mu_f^{(0)}$ and $\upsilon_b^{(0)}$ both corresponding to the BM transport based on the given coupling: $I^{(0)} = M^{C_{0,1}}$, $F^{(0)} = B^{(0)} = I^{(0)}$. Then, looking separately at either of the sequences $F^{(i)}$, $i \geq 1$, and $B^{(i)}$, $i \geq 1$, we obtain that $F^{(i)} = B^{(i)} = I^{(i)}$, $i \geq 1$. $\qquad\square$

**Lemma 2** (Loss Interpretation)**.** *It holds that*

$$
\begin{aligned}
\mathbb{KL}(B_0 \parallel \Psi_0) + \mathbb{L}_f(\mu_f; \upsilon_b) &= \mathbb{KL}(\Pi^{B_{0,1}} \parallel F) + C_1(B) = \mathbb{KL}(M^{B_{0,1}} \parallel F) + C_2(B), \\
\mathbb{KL}(F_1 \parallel \Psi_1) + \mathbb{L}_b(\upsilon_b; \mu_f) &= \mathbb{KL}(\Pi^{F_{0,1}} \parallel B) + D_1(F) = \mathbb{KL}(M^{F_{0,1}} \parallel B) + D_2(F),
\end{aligned}
\tag{15}
$$

*for $C_1(B), C_2(B)$ independent of $F$, $D_1(F), D_2(F)$ independent of $B$, with $0 \leq C_1(B) \leq C_2(B)$ and $0 \leq D_1(F) \leq D_2(F)$.*

*Proof.* We consider only $\mathbb{L}_f(\mu_f; \upsilon_b)$, the arguments for $\mathbb{L}_b(\upsilon_b; \mu_f)$ are symmetric. By Girsanov Theorem (Øksendal, 2013) and by the marginal-conditional decomposition of Kullback-Leibler divergences we have

$$
\mathbb{KL}(M^{B_{0,1}} \parallel F) = \mathbb{KL}(B_0 \parallel \Psi_0) + \mathop{\mathbb{E}}_{\Pi^{B_{0,1}}} \left[ \frac{1}{2} \int_0^1 \|\mu_f(X_t, t) - \mu_m^{B_{0,1}}(X_t, t)\|^2 dt \right],
$$

$$
\mathbb{KL}(\Pi^{B_{0,1}} \parallel F) = \mathbb{KL}(B_0 \parallel \Psi_0) + \mathop{\mathbb{E}}_{\Pi^{B_{0,1}}} \left[ \frac{1}{2} \int_0^1 \|\mu_f(X_t, t) - \mu_\pi^{B_{0,1}}(X_t, t, X_0)\|^2 dt \right],
$$

$$
\mathbb{KL}(\Pi^{B_{0,1}} \parallel \mathcal{F}) = \mathbb{KL}(B_0 \parallel \Psi_0) + \mathop{\mathbb{E}}_{\Pi^{B_{0,1}}} \left[ \frac{1}{2} \int_0^1 \|\mu_f(X_t, t) - \mu_{01}(X_t, t, X_1)\|^2 dt \right]
$$

$$
= \mathbb{KL}(B_0 \parallel \Psi_0) + \mathbb{L}_f(\mu_f; \upsilon_b),
$$

where $\mu_\pi^{B_{0,1}}(X_t, t, X_0) := \mathbb{E}_{\Pi^{B_{0,1}}}[\mu_{01}(X_t, t, X_1)|X_t, X_0]$, $\mu_m^{B_{0,1}}(X_t, t) := \mathbb{E}_{\Pi^{B_{0,1}}}[\mu_{01}(X_t, t, X_1)|X_t]$, and $\mathcal{F}$ is distribution of the non-Markov diffusion solution to the auxiliary SDE

$$
X_0 \sim \Psi_0, \quad dX_t = [\mu_f(X_t, t) - \mu_{01}(X_t, t, X_1) + \mu_\pi^{B_{0,1}}(X_t, t, X_0)]dt + \sigma dW_t, \quad t \in [0, 1]. \tag{$\mathcal{F}$}
$$

By the tower property of conditional expectations and by the conditional Jensen inequality it follows that

$$
\mathbb{KL}(\Pi^{B_{0,1}} \parallel \mathcal{F}) - \mathbb{KL}(\Pi^{B_{0,1}} \parallel F)
$$

$$
= \mathop{\mathbb{E}}_{\Pi^{B_{0,1}}} \left[ \frac{1}{2} \int_0^1 \|\mu_f(X_t, t) - \mu_{01}(X_t, t, X_1)\|^2 - \|\mu_f(X_t, t) - \mu_\pi^{B_{0,1}}(X_t, t, X_0)\|^2 dt \right]
$$

$$
= \mathop{\mathbb{E}}_{\Pi^{B_{0,1}}} \left[ \frac{1}{2} \int_0^1 \|\mu_{01}(X_t, t, X_1)\|^2 - \|\mu_\pi^{B_{0,1}}(X_t, t, X_0)\|^2 dt \right] = C_1(B) \geq 0.
$$

By the Pythagorean property of the BM transport (Liu et al., 2022; Peluchetti, 2023)

$$
\mathbb{KL}(\Pi^{B_{0,1}} \parallel F) - \mathbb{KL}(M^{B_{0,1}} \parallel F) = \mathbb{KL}(\Pi^{B_{0,1}} \parallel M^{B_{0,1}}) = K(B) \geq 0.
$$

Taking $C_2(B) = C_1(B) + K(B)$ completes the proof. $\qquad\square$

**Lemma 3** ($\mathcal{R}$-stability of $F^{(\lambda)}, B^{(\lambda)}$)**.** *If $F', B' \in \mathcal{R}$, then $F^{(\lambda)}, B^{(\lambda)} \in \mathcal{R}$ for each $\lambda \in [0, 1]$.*

*Proof.* By the marginal-conditional decomposition of Kullback-Leibler divergences

$$
\mathbb{KL}(F \parallel B') = \mathbb{KL}(F_{0,1} \parallel B'_{0,1}) + \mathop{\mathbb{E}}_{F_{0,1}} [\mathbb{KL}(F_{|0,1} \parallel B'_{|0,1})],
$$

$$
\mathbb{KL}(F \parallel F') = \mathbb{KL}(F_{0,1} \parallel F'_{0,1}) + \mathop{\mathbb{E}}_{F_{0,1}} [\mathbb{KL}(F_{|0,1} \parallel F'_{|0,1})],
$$

and $B'_{|0,1} = F'_{|0,1} = R_{|0,1}$, hence

$$F^{(\lambda)} := \underset{F \in \mathcal{P}(\Psi_0, \cdot)}{\arg\min} \ \lambda \mathbb{KL}(F_{0,1} \parallel B'_{0,1}) + (1 - \lambda)\mathbb{KL}(F_{0,1} \parallel F'_{0,1}) + \underset{F_{0,1}}{\mathbb{E}} \left[\mathbb{KL}(F_{|0,1} \parallel R_{|0,1})\right],$$

$$B^{(\lambda)} := \underset{B \in \mathcal{P}(\cdot, \Psi_1)}{\arg\min} \ \lambda \mathbb{KL}(B_{0,1} \parallel F'_{0,1}) + (1 - \lambda)\mathbb{KL}(B_{0,1} \parallel B'_{0,1}) + \underset{B_{0,1}}{\mathbb{E}} \left[\mathbb{KL}(B_{|0,1} \parallel R_{|0,1})\right],$$

and thus $F^{(\lambda)}_{|0,1} = B^{(\lambda)}_{|0,1} = R_{|0,1}$, which completes the proof. $\square$

**Lemma 4** ($\mathcal{S}$-stability of $F^{(\lambda)}, B^{(\lambda)}$)**.** *If $F', B' \in \mathcal{S}$, then $F^{(\lambda)}, B^{(\lambda)} \in \mathcal{S}$ for each $\lambda \in [0, 1]$.*

*Proof.* In view of Lemma 3, we have to verify that $F^{(\lambda)}_{0,1}, B^{(\lambda)}_{0,1}$ solve the EOT problems (3) for some marginal distributions if $F'_{0,1}, B'_{0,1}$ do. For simplicity, we assume that all of $F^{(\lambda)}_{0,1}, B^{(\lambda)}_{0,1}, F'_{0,1}, B'_{0,1}$ admits positive densities on $\mathbb{R}^{d \times d}$, and that $\Psi_0$ and $\Psi_1$ admits positive densities on $\mathbb{R}^d$. The steps of this proof carry over to the more general measure-theoretic setting.

We know that $f^{(\lambda)}_{0,1}(x_0, x_1) = \psi_0(x_0) f^{(\lambda)}_{1|0}(x_1|x_0)$ and $b^{(\lambda)}_{0,1}(x_0, x_1) = b^{(\lambda)}_{0|1}(x_0|x_1)\psi_1(x_1)$, where $f^{(\lambda)}_{1|0}(x_1|x_0) \propto b'_{1|0}(x_1|x_0)^\lambda f'_{1|0}(x_1|x_0)^{1-\lambda}$ and $b^{(\lambda)}_{0|1}(x_0|x_1) \propto f'_{0|1}(x_0|x_1)^\lambda b'_{0|1}(x_0|x_1)^{1-\lambda}$ (see Section 4.2). On the other hand

$$f'_{0,1}(x_0, x_1) = \exp\left\{\phi^{f'}_0(x_0) + \phi^{f'}_1(x_1) - \frac{\kappa(x_0, x_1)}{\varepsilon}\right\},$$

$$b'_{0,1}(x_0, x_1) = \exp\left\{\phi^{b'}_0(x_0) + \phi^{b'}_1(x_1) - \frac{\kappa(x_0, x_1)}{\varepsilon}\right\},$$

for the Schrödinger potentials[2] $\phi^{f'}_0(x_0), \phi^{f'}_1(x_1)$ and $\phi^{b'}_0(x_0) + \phi^{b'}_1(x_1)$ (Léonard, 2014a). It follows by direct computation that $f^{(\lambda)}_{0,1}(x_0, x_1)$ and $b^{(\lambda)}_{0,1}(x_0, x_1)$ satisfy:

$$f^{(\lambda)}_{0,1}(x_0, x_1) = \exp\left\{\phi^{f,\lambda}_0(x_0) + \phi^{f,\lambda}_1(x_1) - \frac{\kappa(x_0, x_1)}{\varepsilon}\right\},$$

$$b^{(\lambda)}_{0,1}(x_0, x_1) = \exp\left\{\phi^{b,\lambda}_0(x_0) + \phi^{f,\lambda}_1(x_1) - \frac{\kappa(x_0, x_1)}{\varepsilon}\right\},$$

for some other Schrödinger potentials $\phi^{f,\lambda}_0(x_0), \phi^{f,\lambda}_1(x_1)$ and $\phi^{b,\lambda}_0(x_0), \phi^{f,\lambda}_1(x_1)$. $\square$

## C Additional Results

### C.1 Infinitesimal Minimization, Gaussian Case

Consider the one-dimensional case $d = 1$, with target Gaussian marginal distributions $\Psi_0 = \mathcal{N}(\mu_0, \sigma_0^2)$ and $\Psi_1 = \mathcal{N}(\mu_1, \sigma_1^2)$, and a reference diffusion distribution $R$ associated with ($R$). In this setting, the solution to the static Schrödinger bridge problem (2) is known analytically and is given by a bivariate Gaussian distribution (Mallasto et al., 2022).

We hypothesize that conditional Gaussian densities for $f^{(l)}_{1|0}(x_1|x_0)$ and $b^{(l)}_{0|1}(x_0|x_1)$ solve (19). Specifically, we propose $F^{(l)}_{1|0} = \mathcal{N}(A^f_l x_0 + a^f_l, v^f_l)$ and $B^{(l)}_{0|1} = \mathcal{N}(A^b_l x_1 + a^b_l, v^b_l)$, where $A^f_l, a^f_l, A^b_l, a^b_l \in \mathbb{R}$ and $v^f_l, v^b_l \in \mathbb{R}_{>0}$ are algorithmic-time dependent scalar parameters. By construction, $F^{(l)}_0 = \mathcal{N}(\mu_0, \sigma_0^2)$ and $B^{(l)}_1 = \mathcal{N}(\mu_1, \sigma_1^2)$ for each $l \geq 0$. Substituting these expressions for $f^{(l)}_{1|0}(x_1|x_0)$ and $b^{(l)}_{0|1}(x_0|x_1)$ into (19) yields a six-dimensional ODE system in the parameters. The initial conditions are $A^b_0 = A^f_0 = 1$, $a^f_0 = a^b_0 = 0$, $v^f_0 = v^b_0 = \sigma^2$, corresponding to initial null drift terms for ($F(\theta), \overleftarrow{B}(\theta)$), as discussed in Section 4.2.

To numerically solve the ODE and determine the values of $A^f_l, a^f_l, v^f_l, A^b_l, a^b_l, v^b_l$ over $l \in [0, L]$ for some $L > 0$, we evaluate (19) for three different pairs of $(x_0, x_1)$. This provides sufficient constraints to identify the

---

[2]We formulate the potential with respect to the Lebesgue measure on $\mathbb{R}^d$.

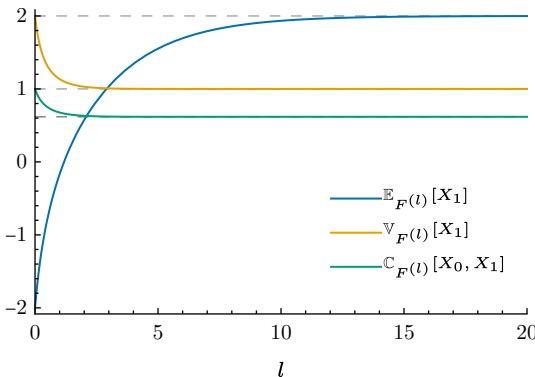

Figure 2: Algorithmic-time $l$ evolution of $\mathbb{E}_{F^{(l)}}[X_1]$, $\mathbb{V}_{F^{(l)}}[X_1]$, $\mathbb{C}_{F^{(l)}}[X_0, X_1]$, compared with $\mathbb{E}_S[X_1]$, $\mathbb{V}_S[X_1]$, $\mathbb{C}_S[X_0, X_1]$ as dashed gray lines.

parameters. Subsequently, we verify that the proposed functional forms for $f_{1|0}^{(l)}(x_1|x_0)$ and $b_{0|1}^{(l)}(x_0|x_1)$ indeed solve (19).

We examine the scenario where $\mu_0 = -2, \mu_1 = 2, \sigma_0 = \sigma_1 = \sigma = 1$. Figure 2 illustrates the evolution of $\mathbb{E}_{F^{(l)}}[X_1]$, $\mathbb{V}_{F^{(l)}}[X_1]$, and $\mathbb{C}_{F^{(l)}}[X_0, X_1]$ over algorithmic time $l$. These quantities represent the mean and variance of $X_1$ and the covariance between $X_0$ and $X_1$ according to $F^{(l)}$, respectively. The corresponding values $\mathbb{E}_S[X_1]$, $\mathbb{V}_S[X_1]$, and $\mathbb{C}_S[X_0, X_1]$ for the static Schrödinger bridge solution $S_{0,1}$ from Mallasto et al. (2022) are depicted as dashed gray lines, demonstrating convergence.

## C.2 Results for Additional Discretization Intervals

In Table 3 we report the results for metric (21) obtained by considering the $\mathrm{BM}^2$, I-BM and DIPF methodologies for different discretization intervals $\Delta t = 1/T$ where $T$ is the number of time-steps. For all methods we employ the same number of time-steps at training and inference (testing) time. We recall that in Section 5 200 time-steps have been employed to produce the results of Tables 1 and 2, and that we rely exclusively on the Euler–Maruyama discretization scheme.

| Method | $\varepsilon=0.1$ | | | | $\varepsilon=1$ | | | | $\varepsilon=10$ | | | |
| | $d=2$ | $d=16$ | $d=64$ | $d=128$ | $d=2$ | $d=16$ | $d=64$ | $d=128$ | $d=2$ | $d=16$ | $d=64$ | $d=128$ |
|---|---|---|---|---|---|---|---|---|---|---|---|---|
| SB(100) | 0.08 _0.00_ | 0.10 _0.00_ | 0.37 _0.00_ | 0.94 _0.00_ | 0.10 _0.00_ | 0.19 _0.00_ | 0.49 _0.00_ | 1.25 _0.00_ | 0.12 _0.00_ | 2.96 _0.00_ | 14.86 _0.02_ | 28.90 _0.03_ |
| SB(50) | 0.23 _0.00_ | 0.22 _0.00_ | 0.44 _0.00_ | 1.00 _0.00_ | 0.15 _0.00_ | 0.24 _0.00_ | 0.61 _0.00_ | 1.49 _0.00_ | 0.13 _0.00_ | 7.24 _0.01_ | 37.24 _0.04_ | 72.34 _0.06_ |
| $\mathrm{BM}^2$(100) | 1.17 _0.79_ | 5.08 _0.22_ | 7.12 _0.71_ | 7.94 _0.78_ | 0.14 _0.04_ | 0.41 _0.03_ | 1.71 _0.14_ | 8.91 _0.35_ | 0.15 _0.02_ | 4.06 _0.03_ | 55.10 _1.13_ | 295.82 _10.60_ |
| $\mathrm{BM}^2$(50) | 0.80 _0.52_ | 4.59 _0.41_ | 6.46 _0.87_ | 8.47 _0.73_ | 0.21 _0.09_ | 0.48 _0.03_ | 1.85 _0.10_ | 9.57 _0.27_ | 0.16 _0.01_ | 8.17 _0.04_ | 78.76 _1.85_ | 202.22 _41.38_ |
| DIPF(100) | 7.28 _4.19_ | 15.72 _1.31_ | 16.44 _0.89_ | 26.69 _0.71_ | 1.43 _0.45_ | 5.96 _0.32_ | 11.98 _0.82_ | 22.60 _0.86_ | 1.14 _0.07_ | 7.36 _0.16_ | 85.42 _2.64_ | 212.19 _1.19_ |
| DIPF(50) | 6.90 _3.01_ | 12.37 _1.40_ | 14.98 _2.16_ | 26.91 _2.71_ | 1.44 _0.33_ | 5.09 _0.43_ | 10.73 _0.29_ | 20.20 _0.93_ | 2.18 _0.06_ | 11.32 _0.18_ | 75.80 _1.02_ | 226.81 _0.92_ |
| I-BM(100) | 0.98 _0.22_ | 4.54 _0.98_ | 6.35 _0.84_ | 13.60 _1.23_ | 0.18 _0.04_ | 0.51 _0.03_ | 2.09 _0.23_ | 7.30 _1.06_ | 0.15 _0.02_ | 5.95 _0.10_ | 113.91 _0.93_ | 494.32 _3.99_ |
| I-BM(50) | 1.58 _0.61_ | 4.21 _0.20_ | 6.35 _1.01_ | 13.75 _2.93_ | 0.28 _0.05_ | 0.60 _0.04_ | 1.96 _0.25_ | 7.31 _1.24_ | 0.16 _0.02_ | 9.14 _0.07_ | 114.40 _0.89_ | 432.20 _5.82_ |

Table 3: Results analogous to Table 2 but for varying discretization intervals $\Delta t = 1/T$, where the number of time-steps $T$ is indicated in parentheses after each method name.

## D  Python Code

```python
1  # dimensions: B: batch; D: data; T: time_steps + 1
2  # required: sample_0(batch_dim, device), sample_1(batch_dim, device), fwd_drift_fn(x, t), bwd_drift_fn(x, t)
3  import torch as th
4
5  # sampling from R_{t|0,1} (5): (B, D), (B, D), (B,), () -> (B, D)
6  def sample_bridge(x_0, x_1, t, sigma):
7      B, D = x_0.shape
8      mean_t = (1 - t[..., None]) * x_0 + t[..., None] * x_1   # (B, D)
9      var_t = sigma**2 * t[..., None] * (1 - t[..., None])     # (B, D)
10     z_t = th.randn_like(x_0)                                 # (B, D)
11     x_t = mean_t + th.sqrt(var_t) * z_t                      # (B, D)
12     return x_t
13
14 # fwd BM target (6): (B, D), (B, D), (B,) -> (B, D)
15 def fwd_target(x_t, x_1, t):
16     return (x_1 - x_t) / (1 - t[..., None])  # (B, D)
17
18 # fwd BM target (7): (B, D), (B, D), (B,) -> (B, D)
19 def bwd_target(x_t, x_0, t):
20     return (x_0 - x_t) / t[..., None]  # (B, D)
21
22 # Euler-Maruyama discretization scheme: fn(x, t), (B, D), (T), () -> (B, D)
23 def discretization(drift_fn, initial_value, times, sigma):
24     B, D = initial_value.shape
25     times = times[..., None].expand(-1, B)                            # (T, B)
26     x_prev_t = initial_value                                         # (B, D)
27     for prev_t, t in zip(times[:-1], times[1:]):                     # (B), (B)
28         dt = t - prev_t                                             # (B)
29         drift_t = drift_fn(x_prev_t, prev_t)                        # (B, D)
30         drift_part_t = drift_t * dt[..., None]                      # (B, D)
31         eps_t = th.randn_like(x_prev_t)                             # (B, D)
32         diffusion_part_t = (sigma * th.sqrt(th.abs(dt)))[..., None] * eps_t  # (B, D)
33         x_t = x_prev_t + drift_part_t + diffusion_part_t           # (B, D)
34         x_prev_t = x_t                                             # (B, D)
35     return x_t
36
37 # BM^2 loss computation: fn(b, d), fn(b, d), fn(x, t), fn(x, t), (), (), (), () -> ()
38 def sample_loss(sample_0, sample_1, fwd_drift_fn, bwd_drift_fn, batch_dim, time_steps, sigma, device):
39     # sample from the target marginals:
40     f_0 = sample_0(batch_dim, device)                                # (B, D)
41     b_1 = sample_1(batch_dim, device)                                # (B, D)
42     # sample according to current (F(θ)) and (B⃖(θ)):
43     fwd_times = th.linspace(0.0, 1.0, time_steps + 1, device=device)  # [0, 1/time_steps, ..., 1]
44     bwd_times = th.linspace(1.0, 0.0, time_steps + 1, device=device)  # [1, ..., 1/time_steps, 0]
45     f_1 = discretization(fwd_drift_fn, f_0, fwd_times, sigma).detach()  # (B, D)
46     b_0 = discretization(bwd_drift_fn, b_1, bwd_times, sigma).detach()  # (B, D)
47     # sample time and mixture processes based on F_{0,1}(θ) and B_{0,1}(θ):
48     t = th.rand((batch_dim,), device=device)                        # (B)
49     pi_f_t = sample_bridge(f_0, f_1, t, sigma)                      # (B, D)
50     pi_b_t = sample_bridge(b_0, b_1, t, sigma)                      # (B, D)
51     # define regression targets and model predictions:
52     target_f_t = fwd_target(pi_b_t, b_1, t)                         # (B, D)
53     target_b_t = bwd_target(pi_f_t, f_0, t)                         # (B, D)
54     prediction_f_t = fwd_drift_fn(pi_b_t, t)                        # (B, D)
55     prediction_b_t = bwd_drift_fn(pi_f_t, t)                        # (B, D)
56     # compute loss:
57     loss_f_t = th.sum((target_f_t - prediction_f_t)**2, dim=1) / 2  # (B)
58     loss_b_t = th.sum((target_b_t - prediction_b_t)**2, dim=1) / 2  # (B)
59     loss_t = th.mean(loss_f_t + loss_b_t)                          # ()
60     return loss_t
```

Listing 1: Basic implementation of BM$^2$ loss computation (Algorithm 1) in PyTorch.

