# OpenReview forum: "BM$^2$: Coupled Schrödinger Bridge Matching"
_TMLR — Accepted by TMLR_

### Review · Reviewer_BMAP · 2024-10-18

**Summary Of Contributions:**

The paper presents a new method called Coupled Bridge Matching ($BM^2$), which addresses the Schrödinger Bridge problem and provides theoretical analysis and empirical results to support the effectiveness of this approach. The core idea is to jointly and simultaneously (rather than indipendently) learn the forward and the backward dynamics of the Schrödinger Bridge and to train a single set of neural networks in one optimization loop via stochastic gradient descent (SGD).

**Audience:**

Yes

**Broader Impact Concerns:**

There are no concerns on the ethical implications of the work.

**Claims And Evidence:**

Yes

**Requested Changes:**

While the paper makes significant strides, a few aspects could benefit from further exploration. Look at the *Weaknesses* section.

**Strengths And Weaknesses:**

*Strenghts*
1. *Joint learning*. Unlike Iterative Bridge Matching (I-BM) and Diffusion Schrödinger Bridge (DSB), where the forward and backward processes are learned separately in an alternating manner (training the forward process first, then the backward process, and repeating the cycle), $BM^2$ learns these two processes jointly. By coupling the forward and backward dynamics during training, $BM^2$ reduces overall approximation errors. This coupling allows the two processes to mutually inform and constrain each other, effectively acting as checks on one another, leading to more consistent and accurate results.
2. *Single optimization loop.* Unlike I-BM and DSB, which require an iterative refinement process where each iteration updates the drift functions based on progressively refined estimates of the true solution, $BM^2$ bypasses this need for iteration. Instead, $BM^2$ learns the optimal drift functions in a single optimization loop by directly training a neural network to find the right parameters, using stochastic gradient descent (SGD). This process only involves Monte Carlo sampling of the initial and terminal states, without the need to iteratively re-compute or refine the drift at each step.

*Weaknesses*
1. *Convergence guarantees.* The paper offers theoretical analysis, but its theoretical convergence is not as rigorously guaranteed as in iterative methods like I-BM and DSB. Specifically, Theorem 2 in the paper suggests that if $BM^2$ reaches a point where further reduction in the loss function is not possible and if the forward and backward processes are the same, then the solution matches the true Schrödinger bridge. This gives some assurance of convergence. However, this result doesn't provide a complete convergence proof in the way iterative methods do. The connection between reductions in KL divergence/loss functions during training and overall convergence to the true Schrödinger bridge remains unclear. The guarantee is more empirical and relies on specific conditions being met during the optimization process, such as the neural network’s ability to accurately approximate the drift functions and the avoidance of issues like local minima.
2.  *Practical limitations*. In practice, the method’s performance could be limited by the discretization of the underlying stochastic differential equations (SDEs) and the quality of neural network approximations. These limitations are acknowledged but not thoroughly quantified in the paper.
3. *Computational efficiency*. While $BM^2$ is theoretically more efficient due to its non-iterative design, in practice, its reliance on neural networks and stochastic gradient descent (SGD) introduces potential computational challenges. First, $BM^2$ is subject to common neural network limitations, such as overfitting, vanishing or exploding gradients, and sensitivity to network architecture. If the neural network fails to accurately model the dynamics of the underlying stochastic differential equation (SDE), $BM^2$'s performance may suffer. Additionally, $BM^2$’s training involves learning drift functions for both forward and backward processes simultaneously, which can become computationally demanding—especially as the problem’s dimensionality increases or the neural network architecture becomes more complex.
3. *Empirical breadth*. The empirical results are promising but focus heavily on specific settings.

---

> ### Author Response · Authors · 2024-11-10
> **Response to Reviewer BMAP**
>
> > Convergence guarantees. The paper offers theoretical analysis, but its theoretical convergence is not as rigorously guaranteed as in iterative methods like I-BM and DSB [...]
>
> We acknowledge that our theoretical results represent only an initial step toward understanding the convergence properties of BM², particularly since the proposed method more significantly departs from well-established iterative approaches like IPF/Sinkhorn.
>
> > Practical limitations. In practice, the method’s performance could be limited by the discretization of the underlying stochastic differential equations (SDEs) and the quality of neural network approximations. These limitations are acknowledged but not thoroughly quantified in the paper.
>
> Table 2 shows initial results on how the discretization interval affects our method.
> Looking at the first row (EOT) which only accounts for Monte Carlo estimation error, versus the second row (SB(Discr.)) which includes SDE discretization error, we see that using 200 time steps works well up to $ε=1$ but falls short at $ε=10$.
> We will expand this analysis in the revised manuscript by testing more discretization intervals across all methods and update the results of Section 5 accordingly.
>
> Since we have access to the oracle target drift in our benchmark, we can evaluate how well the neural network approximates it using Table 1 (Equation (18)).
> Performance drops at very high $ε$ values - though these cases are less practically relevant since the optimal coupling becomes nearly independent.
> More importantly, performance also degrades at low $ε$ values, which are of interest when approximating the (unregularized) optimal transport map is the goal.
> This is a common challenge for all methods, particularly in high dimensions, and in our response to Reviewer Jqq8 we discuss potential improvements there.
>
> > Computational efficiency. While BM² is theoretically more efficient due to its non-iterative design, in practice, its reliance on neural networks and stochastic gradient descent (SGD) introduces potential computational challenges. First, BM² is subject to common neural network limitations, such as overfitting, vanishing or exploding gradients, and sensitivity to network architecture. If the neural network fails to accurately model the dynamics of the underlying stochastic differential equation (SDE), BM²'s performance may suffer. Additionally, BM²’s training involves learning drift functions for both forward and backward processes simultaneously, which can become computationally demanding—especially as the problem’s dimensionality increases or the neural network architecture becomes more complex.
>
> We agree that the ability to accurately approximate target drift functions is indeed key for any method addressing the dynamic Schrödinger Bridge problem.
> Using neural networks instead of other approaches like Gaussian Processes (e.g., [8]) gives up some optimization guarantees but often works better in practice on hard problems.
> Our reply to Reviewer Jqq8 on the choice of initial drift function approximators is also relevant in addressing these considerations.
>
> Limited to our benchmark comparisons, we did not specifically observe that joint training of forward and backward drift functions is more challenging than sequential learning, as demonstrated by the results comparing BM² to I-BM in Table 1.
> Previous research has explored joint learning of forward and backward processes even in cases where a single direction would have been sufficient, enabling the introduction of an additional consistency loss (see Appendix G of [6]).
>
> ### References
>
> - [6] Diffusion Schrödinger Bridge Matching
> - [8] Solving Schrödinger Bridges via Maximum Likelihood

---

> > ### Comment · Reviewer_BMAP · 2024-11-15
> >
> > I appreciate the author's comprehensive response. My questions and concerns are sufficiently addressed.

---

### Review · Reviewer_Jqq8 · 2024-10-19

**Summary Of Contributions:**

The paper introduces a novel approach for learning Schrödinger bridges. $BM^2$ addresses the problem of Schrodinger BM transport between two distributions, leveraging a reference diffusion process. Using a coupled system of forward and backward transport equations, it eliminates the need for iterative methods by employing a single optimization loop, using stochastic gradient descent, significantly simplifying the training process. The method is exact in its idealized form, with only neural network approximation and discretization introducing practical errors. The paper provides a preliminary theoretical analysis of $BM^2$ convergence properties, supported by simple numerical experiments that demonstrate its effectiveness against similar methods.

**Audience:**

Yes

**Broader Impact Concerns:**

Sufficiently addressed.

**Claims And Evidence:**

Yes

**Requested Changes:**

Questions and Requested Changes:
- What about incorporating a consistency loss by enforcing $F$ and $\overleftarrow{B}$ to be time-reversals of each other, as done in [1]? While you mention in the conclusion that this is not necessary, could it not potentially speed up the training process?
- Contrary to DIPF and I-BM, the structure of $BM^2$ removes the problem of the initialisation choice for $\mu_f^{(0)}, v_b^{(0)}$ (which is the choice of initial joint distribution for DIPF, for instance). Can you confirm it is true?
- In the experiments, what value of $\sigma$ is used for $BM^2$? What is the EMA rate?
- What is the rationale behind the $BM^2_{\sigma}$ model? What $\sigma$ is used for sampling? Could this lead to a more efficient model for deterministic ODEs as $\sigma \to 0$? Is it valid to set $\sigma = 0$ for sampling?
- Regarding the neural network architecture, you mention using a single network for $\mu_f$ and $v_b$. Does this imply two output channels? How is $\sigma$ incorporated into the architecture— through positional embedding?
- While the paper does not focus on transfer or generation on higher-dimensional datasets (like MNIST or CIFAR10) as stated in the conclusion, it would be beneficial to include some small experiments in this area and compare them to I-BM and DIPF. This would likely enhance the paper's appeal to a broader audience.


[1] Yuyang Shi, Valentin De Bortoli, Andrew Campbell, Arnaud Doucet (2023) -- Diffusion Schrödinger Bridge Matching

**Strengths And Weaknesses:**

Strengths:
- Extremely clear and effective presentation.
- Excellent contextualization within the existing literature and comparison to related methods.
- The introduction of the coupled system is natural, resembling the structure of the EM algorithm.
- Includes I-BM and DIPF as special cases when complete minimization is performed at each step.
- Enables a non-iterative approach, with a single optimization loop instead of the outer and inner loops used in I-BM or DIPF. This partial minimization has strong theoretical backing in Theorem 2, suggesting promising effectiveness.
- Demonstrates superior experimental results.
- Offers further theoretical research opportunities, as highlighted in Section 4.2.3.

Weaknesses:

- Not really, maybe lacking higher dimensional experiments, to show improved generation quality or training speed.

---

> ### Author Response · Authors · 2024-11-10
> **Response to Reviewer Jqq8 (part 1)**
>
> > What about incorporating a consistency loss by enforcing and to be time-reversals of each other, as done in [6]? While you mention in the conclusion that this is not necessary, could it not potentially speed up the training process?
>
> The suggested approach is feasible and can be implemented as follows.
> The learning process comprises two components: a forward diffusion $F$ and a backward diffusion $B$.
> At equilibrium, our objective is to achieve $F = B = Π^{F_{0,1}} = Π^{B_{0,1}}$.
> Nelson's relationship establishes that the sum of forward and backward drifts equals the diffusion squared times the score (under our backward time convention).
> Based on this principle, we can introduce an additional forward-backward consistency loss:
>
> $$
> ℂ_{f,b}(θ;θ') ≔ 𝔼_{\frac{1}{2}(Π^{F_{0,1}}(θ') + Π^{B_{0,1}}(θ'))}\Big[\frac{1}{2}\int_0^1‖μ_f(X_t,t,θ) + υ_b(X_t,t,θ) - γ_{01}(X_t,t,X_0,X_1)‖^2dt\Big],
> $$
>
> where $θ'$ represents an independent copy of $θ$ (implemented using the stop-gradient operator), and
>
> $$
> γ_{01}(X_t,t,X_0,X_1) ≔ σ^2∇_{X_t}\log r_{t|0,1}(X_t|X_0,X_1) = \frac{X_0(1 - t) + X_1 t - X_t}{t(1 - t)}.
> $$
>
> The theoretical foundation for this loss stems from the fact that for any mixture process $Π^{C_{0,1}} = C_{0,1}R_{|0,1}$ it holds that:
>
> $$
> σ^2 ∇\log π_t^{C_{0,1}}(x_t) = 𝔼_{Π^{C_{0,1}}}[γ_{01}(X_t,t,X_0,X_1)|X_t=x_t].
> $$
>
> While this consistency loss bears similarity to that presented on page 33 of [6], our approach differs in a key aspect: rather than using a fixed process $Π$ corresponding to a specific I-BM iteration, we employ the dynamic mixture $\frac{1}{2}(Π^{F_{0,1}} + Π^{B_{0,1}})$.
> Including the consistency loss incurs approximately twice the computational cost per optimization step, as the neural networks $μ_f(X_t,t,θ)$ and $υ_b(X_t,t,θ)$ require evaluation on additional inputs: it does not suffice to reuse the computations required for the estimation of the losses $𝕃_f$ and $𝕃_b$ in Equation (11).
> We will evaluate whether this approach provides competitive performance within the same computational budget and include the findings in the revised manuscript.
>
> > Contrary to DIPF and I-BM, the structure of removes the problem of the initialisation choice for $μ_f^{(0)}$, $υ_b^{(0)}$ (which is the choice of initial joint distribution for DIPF, for instance). Can you confirm it is true?
>
> The choice of initial values for the drifts $μ_f(x,t,θ)$ and $υ_b(x,t,θ)$ is flexible in principle.
> In our numerical experiments, we found that initializing them to zero worked well, and we will clarify this in the revised manuscript.
>
> A key challenge that can emerge during early training is the simulation-inference mismatch.
> For accurate results, the drifts must be learned properly in the regions where the SDE will be simulated.
> When the neural networks approximating these drifts are trained on samples that inadequately represent these regions, we end up extrapolating rather than interpolating, potentially leading to poor approximations.
> This issue is discussed in detail in Sections 2.3 and 6.2 of [4] in the context of DIPF.
>
> Given that processes $F$ and $B$ start quite different from each other, we found that the learning process can be improved by first learning the BM transport in both directions based on the independent coupling $Ψ_0×Ψ_1$.
> The BM transport avoids the simulation-inference mismatch by construction.
> The BM² transport can then be learned using the BM solution as initialization.
> We observed that this strategy enhanced convergence in the EOT benchmark, particularly for low values of $σ$ where the inference-simulation mismatch is most pronounced.
> While we did not mention this in the manuscript since the improvements were modest for our specific benchmark, we will update it to highlight these aspects and suggest this BM transport initialization approach for cases where convergence proves challenging.
>
> > In the experiments, what value of $σ$ is used for BM^2? What is the EMA rate?
>
> In all our experiments, we set $σ = \sqrt{ε}$ to solve the EOT problem with regularization parameter $ε$, which corresponds to the SB problem with diffusion $σ$ (see discussion after Equation (3)).
>
> We consistently used an EMA rate of 0.99 across all experiments.
> We will include this detail in the revised manuscript.

---

> > ### Author Response · Authors · 2024-11-10
> > **Response to Reviewer Jqq8 (part 2)**
> >
> > > What is the rationale behind the model BM $^2_σ$? What $σ$ is used for sampling? Could this lead to a more efficient model for deterministic ODEs as $σ → 0$? Is it valid to set $σ=0$ for sampling?
> >
> > The $BM^2_σ$ model serves as a proof of concept for simultaneously learning multiple couplings.
> > Since the learning problem becomes more challenging as $σ → 0$, as demonstrated in Table 2, we hypothesized that learning the optimal drift across multiple values of $σ$ would enable better adaptation to low $σ$ values through extrapolation based on continuity principles.
> >
> > Furthermore, it is established that when $σ=0$, where BM reduces to the rectified flow, the OT optimal coupling is not generally recovered, as shown in [5].
> > However, the OT solution should theoretically be recoverable through continuity arguments given a continuum of solutions over varying levels of $σ$ approaching zero, as $σ → 0$.
> > This would ultimately yield a deterministic ODE in the limit.
> >
> > Although our reported results are not encouraging, identifying the specific cause requires additional investigation.
> > Several factors could be responsible: model capacity limitations, the (lack of) parameterization of $σ$, or the need for enhanced extrapolation through a PINN-inspired (physics informed NN) approach.
> >
> > For the $BM^2_σ$ implementation, we sample $σ$ from a uniform distribution ranging from 0.1 to 4, as noted in the manuscript.
> >
> > While the BM² algorithm remains mathematically well-defined at $σ=0$, convergence is not guaranteed in this case.
> > In the idealized setting and under some assumptions, BM² corresponds to IPF (Theorem 1), which notably fails to converge when $σ=0$.
> > As previously mentioned, from an implementation standpoint, BM² performance deteriorates significantly at very low $σ$ values.
> > The consistency loss discussed above could prove beneficial in these scenarios.
> >
> > > Regarding the neural network architecture, you mention using a single network for $μ_f$ and $υ_b$. Does this imply two output channels? How is $σ$ incorporated into the architecture— through positional embedding?
> >
> > This is correct, our network outputs $2 * d$ channels, matching the dimensionality $d$ of the problem, we will make this clearer in the revised manuscript.
> > Both $σ$ and $t$ are fed directly into the network as input parameters, so using Fourier embeddings with proper scaling could potentially enhance the algorithm's performance.
> >
> > ### References:
> >
> > - [4] Diffusion Bridge Mixture Transports, Schrödinger Bridge Problems and Generative Modeling
> > - [5] Rectified Flow: A Marginal Preserving Approach to Optimal Transport
> > - [6] Diffusion Schrödinger Bridge Matching

---

> > > ### Comment · Reviewer_Jqq8 · 2024-11-18
> > > **Answer to Rebuttal**
> > >
> > > I thank the authors for their comprehensive answer. My questions have been sufficiently addressed and I greatly appreciate the work outlined to integrate a consistency loss. I will be happy to recommend acceptance of the paper.

---

### Review · Reviewer_ezyt · 2024-10-31

**Summary Of Contributions:**

This paper introduced a new approach to approximating a solution of Schrodinger bridge problem (SB) with neural networks. SB is a problem that is well developed theoretically in control and measure transport theory, but it is well-known that there does not exist a closed-form solution for general setting. The bridge matching (BM) framework (Peluchetti 2021; Shi et al. 2023) offered a new scalable solver for the SB problem, but is still considered to be costly as training requires solving an iteration of the optimization problem (a process called iterated bridge matching) that updates two neural networks sequentially (forward and backward drifts). This work extended the BM framework and propose a a new training scheme that is simpler: coupled both the forward and backward drift matching into a single loss. Some theoretical investigations are shown to analyze the equivalence of BM^2 with previous frameworks, along with empirical demonstrations to show efficiency of the new training loss.

**Audience:**

Yes

**Broader Impact Concerns:**

No.

**Claims And Evidence:**

Yes

**Requested Changes:**

See some of the questions on the experimental side in the Weaknesses section.

**Strengths And Weaknesses:**

Strengths:

- Well-motivated problem and solution. The paper is well-written with a comprehensive literature review and background sections.
- The derivation of the new loss is based on rigorous theoretical argument. The proofs in the Appendix regarding convergence analysis appear to be correct.
- Some simple numerical schemes showing the method's competitive performance against Bridge Matching.

Weaknesses: (mostly on the experimental side)

- The experiments lack some of the recent baselines -- I was surprised to see a detailed discussion of these baselines in Section 6 (for example the strong baseline lightSB), but not inside the numerical experiments itself.
- While the bridge estimation tasks and the KL/BW2-UVP metrics show competitive performance, it is unclear on the benefit of the new loss in the training process itself. It would be welcome if the author could show a simple plot demonstrating normalized training loss of BM^2 compared with BM/DIPF.
 - In a training framework that requires a single neural network to learn both forward and backward drifts, it is intriguing to see how well BM^2 can scale beyond learning bridges between Gaussian/mixture of Gaussians; especially on the stability of the training. My opinion is that the current set of experiments is not large enough to make a good conclusion. However, this is a minor concern, provided that the paper's goal is to introduce a new training regime for diffusion SB and analyze it theoretically.

---

> ### Author Response · Authors · 2024-11-10
> **Response to Reviewer ezyt**
>
> > The experiments lack some of the recent baselines -- I was surprised to see a detailed discussion of these baselines in Section 6 (for example the strong baseline lightSB), but not inside the numerical experiments itself.
>
> A key consideration in comparing with LightSB pertains to the EOT benchmark used in our experiments ([3]).
> This benchmark defines optimal target couplings $π^*_{0,1}$ where $π^*_{1|0}$ is specifically constructed as a mixture of Gaussian distributions (as detailed in Section 5 and Proposition 3.3 of [3]).
>
> Both implementations of LightSB [1,2] employ an identical parametrization approach for their learnable coupling $π^θ_{1|0}$, modeling it as a mixture of Gaussian distributions.
> Consequently, when using five or more Gaussian distributions—matching the fixed number used to define the mixture distribution $π^*_{0,1}$ in the EOT benchmark [3]—the optimal coupling can be easily recovered by construction.
>
> This inherent advantage is acknowledged by the authors of [1], who state: "As clearly seen, our solver outperforms the best solver by a considerable margin. This is reasonable as the benchmark distributions are constructed using the similar principles which our solver exploits, namely, the sum-exp (Gaussian mixture) parameterization of the Schrodinger potential. Therefore, our light solver has a considerable inductive bias for solving the benchmark."
>
> Nonetheless, our results presented in Table 2 are directly comparable to those reported in Table 2 of [1] and Table 1 of [2], as we utilize the identical benchmark with matching parameter choices for $d$ and $ε$, where strong performance is indeed demonstrated.
> In the revised manuscript we will direct the readers to these results.
>
> Regarding the Forward-Backward SB SDE approach proposed in [7], its iterative implementation closely resembles the DIPF, see Appendix C of [7], with the main difference being the use of divergence-based objectives.
> The experimental results in [6] and our own findings in Tables 1 and 2 suggest that bridge matching approaches typically outperform DIPF-like methods in practice.
> The non-iterative variant requires differentiation through simulated paths, resulting in substantially higher computational and memory requirements, which complicates direct comparison with other methods (all currently included methods operate under similar computational constraints and number of SGD iterates).
> Given these considerations, we do not plan to incorporate a comparison with [7], please advise if this decision is acceptable.
>
> > It would be welcome if the author could show a simple plot demonstrating normalized training loss of BM^2 compared with BM/DIPF
>
> The training objectives have different scales: in our experiments we employed the drift matching estimator for DIPF and the exact drift estimator derived from (11) for I-BM and BM² (under different distributions through the training).
> Neither loss function achieves a zero minimum value, and normalizing them to a comparable interpretable scale presents significant challenges.
> Instead, we will include in the revised manuscript a figure demonstrating the convergence behavior of both metrics considered the experiments as a function of SGD iterations across all evaluated methods.
>
> ### References:
>
> - [1] Light Schrödinger Bridge
> - [2] Light and Optimal Schrödinger Bridge Matching
> - [3] Building the Bridge of Schrödinger: A Continuous Entropic Optimal Transport Benchmark
> - [6] Diffusion Schrödinger Bridge Matching
> - [7] Likelihood Training of Schrödinger Bridge using Forward-Backward SDEs Theory

---

> > ### Comment · Reviewer_ezyt · 2024-11-30
> >
> > Really sorry for the late response. The authors' rebuttal has addressed my concern. I recommend acceptance of this excellent work.

---

### Author Response · Authors · 2024-11-10
**Common Response**

We thank the three anonymous reviewers for their thoughtful comments and careful review of our submission.

In this response, we address the common observation that the work would benefit from a larger set of numerical experiments, and summarize the main changes we propose for the revised version of our manuscript.

We address reviewer-specific concerns via individual replies.

### On further numerical experiments

While we agree with the reviewers that a more thorough assessment of the proposed method's performance on additional challenging problems would be valuable, we prefer to confine the experimentation in the present manuscript to the considered EOT benchmarks.
Conducting further experiments for image applications, for instance, would necessitate a stronger focus on implementation aspects, such as neural network architecture, scheduling optimization (Appendix A), both parametric and learned, improved handling of the time input through embeddings, and ad-hoc parametrizations of both the neural network approximator and its targets.

These are undoubtedly important topics, and we are actively working on a separate paper which will focus on these aspects.
Therefore, we prefer to maintain the focus of the present paper on introducing the new method, leaving the detailed exploration of applications to forthcoming work, as many of these practical considerations are relevant for BM methodologies in general and are not limited to BM².
Please note however that we are broadening the scope of our current benchmarking experiment (see below).

### Proposed changes

We outline the following key modifications for the revised manuscript:

- Addition of forward-backward consistency loss and evaluation of its effectiveness in the benchmarking experiments
- Mention of two-stage procedure, from BM transport to BM², to help with simulation-inference mismatch
- Additional benchmark experiments to assess the impact of SDE discretization intervals
- Addition of a new figure illustrating the temporal progression of the two considered metrics for all proposed methodologies
- Refinement of the manuscript's presentation to address each specific concern raised by the reviewers

---

### Decision · Action_Editor_VwXC · 2024-12-05

**Recommendation:** Accept as is

**Comment:**

All the reviewers and I found the paper highly interesting. The only notable weakness lies in the experimental section, which focuses solely on simple toy examples. Nevertheless, we agree that the paper introduces sufficiently novel ideas to make its contribution highly valuable.

**Audience:**

This paper proposes a novel approach to solving the SV problem, a topic of significant current interest within the community, particularly among TMLR's audience.

**Claims And Evidence:**

This paper presents a novel approach to estimating solutions of the Schrödinger Bridge (SB) problem between two distributions $\Psi_0$ and $\Psi_1$. The Bridge Matching (BM) framework (Peluchetti 2021; Shi et al. 2023) offers computational methods for solving the SB problem but suffers from high computational costs. This work extends the BM framework, proposing a simplified training scheme that integrates forward and backward drift matching into a single loss function. Empirical experiments further showcase the efficiency of this new training approach.